# Solving Token Gradient Conflict in Mixture-of-Experts for Large Vision-Language Model

**Longrong Yang[1], Dong Shen[2], Chaoxiang Cai[3], Fan Yang[2], Tingting Gao[2], Di Zhang[2], Xi Li[1,†]**
[1]College of Computer Science and Technology, Zhejiang University
[2]Kuaishou Technology
[3]School of Software Technology, Zhejiang University

## Abstract

The Mixture-of-Experts (MoE) has gained increasing attention in studying Large Vision-Language Models (LVLMs). It uses a sparse model to replace the dense model, achieving comparable performance while activating fewer parameters during inference, thus significantly reducing the inference cost. Existing MoE methods in LVLM encourage different experts to specialize in different tokens, and they usually employ a router to predict the routing of each token. However, the router is not optimized concerning distinct parameter optimization directions generated from tokens within an expert. This may lead to severe interference between tokens within an expert. To address this problem, we propose to use the token-level gradient analysis to **S**olving **T**oken **G**radient **C**onflict (STGC) in this paper. Specifically, we first use token-level gradients to identify *conflicting tokens* in experts. After that, we add a regularization loss tailored to encourage *conflicting tokens* routing from their current experts to other experts, for reducing interference between tokens within an expert. Our method can serve as a plug-in for diverse LVLM methods, and extensive experimental results demonstrate its effectiveness. The code will be publicly available at https://github.com/longrongyang/STGC.

## 1 Introduction

Large Vision-Language Models (LVLMs) have recently demonstrated significant advancements by integrating visual processing modules into Large Language Models (LLMs). Many recent LVLMs (Zhang et al., 2023a; Bai et al., 2023b; Zhang et al., 2023b; Zhao et al., 2023; Chen et al., 2023b) show that large model size and large dataset size are significant to enhance intelligence, *i.e.*, the scaling law. Even when the model size is sufficiently large, models exhibit "Emergent Abilities". Thus, a series of studies (Li et al., 2022; Dai et al., 2023; Liu et al., 2023b) have expanded the model size of LVLMs to 13 billion parameters, leading to state-of-the-art performance on various tasks.

Under realistic applications, deploying such large models requires considerable computational resources, making inference extremely expensive. For reducing the inference cost, a popular solution is using the Mixture-of-Experts (MoE) architecture, replacing the FFN layer with multiple experts, which has been verified by many works (Fedus et al., 2022; Zoph et al., 2022; Komatsuzaki et al., 2022) to achieve comparable performance with dense models when activating fewer parameters.

With multiple experts in the MoE, a fundamental problem is the routing of tokens. To route tokens to different experts, existing MoE works (Lin et al., 2024; Dai et al., 2024) typically train a router, such as a linear layer, to predict the probability of each token dispatched to different experts. The tokens are then dispatched to the experts with the Top-$k$ predicted probability. However, a natural problem arises: *What is the optimization goal of the router to dispatch tokens*?

Given the wide variety of data used in LVLMs, we think that a critical goal of token routing to various experts is to reduce interference between diverse data. Some related LoRA-MoE studies (Chen et al., 2023d; Gou et al., 2023; Shen et al., 2024; Liu & Luo, 2024; Zhou et al., 2024) have also conducted preliminary explorations from this perspective, usually modeling data interference through

---

†Corresponding author is Xi Li.

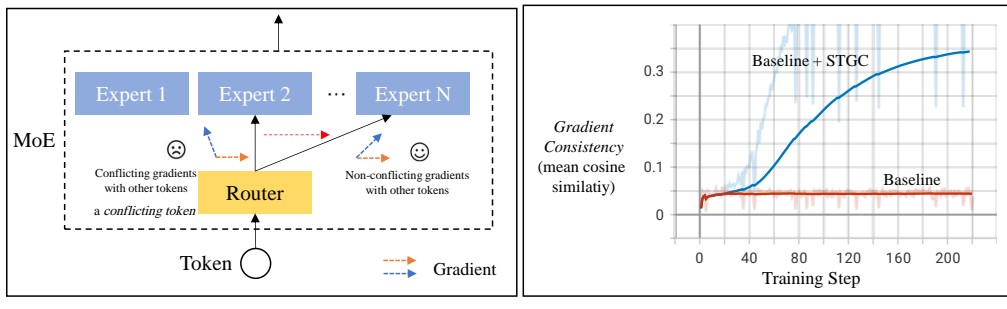

(a) Our goal: reduce gradient conflicts
of tokens within an expert

(b) *Gradient consistency* of tokens within an expert
before and after using STGC

Figure 1: (a) In this work, we aim to solve data interference by adjusting token routing to reduce gradient conflicts. (b) We present statistics regarding *gradient consistency* (the mean cosine similarity between gradients of all tokens within an expert). In experiments, we fed one sample into the LVLM per device for each forward pass. The baseline LVLM is MoE-LLaVA (Lin et al., 2024).

sample-level instruction features or embeddings. For instance, MoCLE (Gou et al., 2023) performs sample-level clustering on instruction embedding, solving data interference by constraining samples of distinct cluster centers to pass different experts. However, even though samples have similar instruction embeddings, they may generate distinct parameter optimization directions due to distinct targets, so embedding-based routing is still risky for optimization interference within an expert. Moreover, since the routing is typically at the token level (Lin et al., 2024; Dai et al., 2024), existing sample-level methods may struggle with interference between tokens (*e.g.*, visual and text tokens) within a sample. Fortunately, the gradient can directly indicate the direction of parameter optimization. Thus, this work aims to model data interference through the lens of **token-level gradients**. As shown in Figure 1 (a), our basic idea is to *optimize the router to reduce gradient conflicts between tokens within an expert*, for solving data interference under complex and real-world scenarios.

To this end, we propose to employ the token-level gradient analysis to design a novel regularization loss for **S**olving **T**oken **G**radient **C**onflict (STGC). The STGC is proposed to answer two key questions: $(i)$ *How to define conflicting tokens?* After forwarding a batch of data, we perform a backward pass to capture the token-level gradients on each expert without updating any model parameters. Within an expert, we compute the average gradient of all tokens, representing the holistic optimization direction of the expert. Then, tokens are identified as *conflicting tokens* if their gradients exhibit a negative cosine similarity to the average gradient. These *conflicting tokens* (outliers) harm the learning of the expert. $(ii)$ *How to solve conflicting tokens?* After identifying *conflicting tokens*, we design a conflict elimination loss to optimize the router to encourage *conflicting tokens* routing away from their current experts. As shown in Figure 1 (b), the STGC enhances gradient consistency between tokens within an expert, *i.e.*, reduces gradient conflicts of tokens.

In conclusion, our contribution can be summarized as:

- Beyond relying on sample-level embedding cues, we propose using token-level gradients to define *conflicting tokens* for modeling data interference in the LVLMs.
- We propose a novel conflict elimination loss to optimize token routing to solve gradient conflicts, making parameter optimization directions generated from tokens within an expert more consistent. This also prompts the further specialization of experts in the MoE.
- Designed as a plug-in, our method can be seamlessly integrated into existing MoE-based LVLMs. Extensive experiments have confirmed its effectiveness.

## 2 RELATED WORKS

### 2.1 LARGE VISION-LANGUAGE MODEL

Large Language Models (LLMs) have demonstrated strong instruction following and generalization capabilities. LLMs can only process textual information, while real-world applications require

models to process visual information, *e.g.*, object detection (Yang et al., 2025) or instance segmentation (Yang et al., 2020; 2022). To incorporate visual information, Large Vision-Language Models (LVLMs) such as GPT-4 and LLaVA utilize frozen visual encoders and trainable visual projectors to integrate visual data into LLMs. Recent works have focused on improving performance through two types of methods. The first type optimizes training strategies, *e.g.*, (Bai et al., 2023b; Chen et al., 2023a). Most works belong to the second type, focusing on enhancing visual components, including expanding visual instruction-tuning datasets (Liu et al., 2023a; Zhang et al., 2023b), improving image encoders (Chen et al., 2023e; Bai et al., 2023b), and aligning the input and projection layers (Lin et al., 2023; Cha et al., 2023; Alayrac et al., 2022; Dai et al., 2023; Ye et al., 2023; Zhao et al., 2023). These efforts, particularly the expansion of visual instruction-tuning datasets and the increase in model scales, have significantly enhanced the visual understanding abilities of LVLMs.

## 2.2 MIXTURE-OF-EXPERTS (MOE)

Efficient training and inference, e.g., (Yang et al., 2023; 2024), are crucial for the deployment of large models. The Mixture-of-Experts (MoE) is a hybrid model consisting of multiple sub-models known as experts and has shown potential in reducing the inference cost (Shazeer et al., 2017). The critical concept of MoE lies in using a router to determine the token set that each expert handles, aiming to reduce interference between tokens from diverse data. Early MoE works have utilized the hard routing mode, where each expert is typically assigned a specific role. For example, a series of works (Bao et al., 2022; Satar et al., 2022; Long et al., 2023; Wang et al., 2022; Shen et al., 2023b) consider vision and language gaps in multi-modal data (Liang et al., 2022), decoupling experts by modal type and assigning a specific role to each expert. The critical property of hard routers is that they eliminate the need to learn the routing. The hard routing has also been widely applied in task-specific MoEs (Kudugunta et al., 2021; Zhu et al., 2022; Li et al., 2023c; Ma et al., 2023).

Then, soft routers enable a dynamic allocation of tokens between different experts. Recent works have mainly focused on soft routers *e.g.*, (Lepikhin et al., 2020; Fedus et al., 2022; Zoph et al., 2022; Komatsuzaki et al., 2022; Shen et al., 2023a; Zadouri et al., 2023; Puigcerver et al., 2023; Chen et al., 2023c; Chalapathi et al., 2024; Zhong et al., 2024). For LVLMs, MoE-LLaVA (Lin et al., 2024), Uni-MoE (Li et al., 2024b), and MoAI (Lee et al., 2024) propose to employ MoE to empower LVLMs. DeepSeekMoE (Dai et al., 2024) and QwenMoE (Bai et al., 2023a) further segment experts by splitting the FFN hidden dimension to achieve further specialization. CuMo (Li et al., 2024a) designs MoE for both the vision encoder and the MLP connector. DYNMOE (Guo et al., 2024) and AdaMoLE (Liu & Luo, 2024) enable each token to determine the number of experts to activate dynamically. Some recent works have claimed that the MoE structure itself is suitable for handling data interference, so they address data interference by adding LoRA-MoE on a fixed FFN (Chen et al., 2023d; Gou et al., 2023; Wu et al., 2024; Chen et al., 2024; Shen et al., 2024; Liu & Luo, 2024; Zhou et al., 2024). LLaVA-MoLE (Chen et al., 2024) resembles MoE-LLaVA, using token representation to predict the routing scores. LoRA-MoE (Chen et al., 2023d) uses instance-level instruction token average representation to predict the routing scores. MoME (Shen et al., 2024) uses instruction embeddings to infer different visual representations' weightings and compute their weighted sum. Then, MoCLE (Gou et al., 2023) clusters instruction embeddings of samples and use the cluster-related learnable embeddings to predict the routing. MoLA (Zhou et al., 2024) constraints the router based on the sample task. These methods usually operate at the sample level, which makes it challenging to address interference between different tokens within a sample. This work utilizes token-level gradients to solve data interference in the MoE.

## 3 METHODOLOGY

### 3.1 OVERVIEW

**Large Vision-Language Model**: A Large Vision-Language Model (LVLM) aims to effectively integrate the capabilities of the pre-trained LLM and a visual model. Specifically, the input of the vision encoder is an image $\mathbf{v} \in \mathbb{R}^{H \times W \times 3}$, where $H$ and $W$ are its height and width, and its output is a visual token sequence $\mathcal{Z} = [z_1, z_2, \cdots, z_P] \in \mathbb{R}^{P \times C}$, where $P$ is the sequence length of visual tokens. Then, a visual projection layer is used to map $\mathcal{Z} \in \mathbb{R}^{P \times C}$ to $\mathcal{V} \in \mathbb{R}^{P \times D}$, where $D$ represents the hidden size of Large Language Model (LLM). Besides, the instruction text is projected as instruction text tokens $\mathcal{T} = [t_1, t_2, \cdots, t_N] \in \mathbb{R}^{N \times D}$, where $N$ represents the sequence length

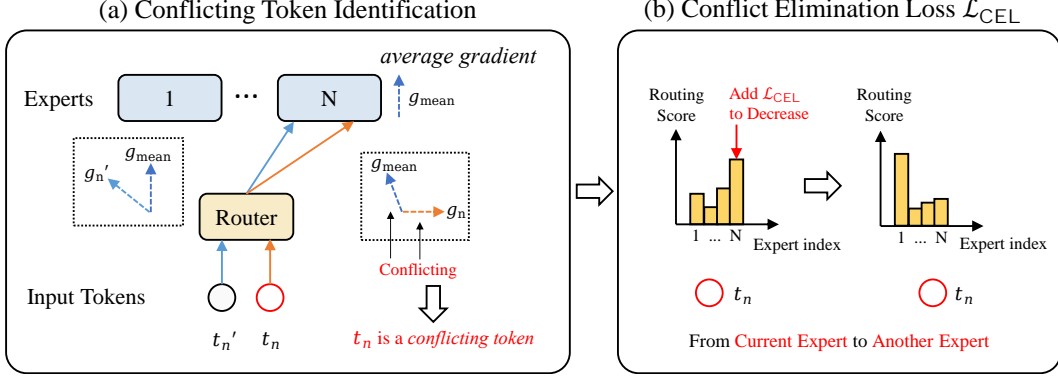

Figure 2: Our pipeline. (a) Conflicting Token Identification. When the gradient of a token has a sufficiently low cosine similarity to the *average gradient* of its assigned expert, this token is marked as a *conflicting token* (an outlier for the expert). (b) Conflict Elimination Loss. We propose a loss aimed at encouraging the routing of *conflicting tokens* from their current experts to other experts.

of instruction text tokens. This model consists of stacked multi-head self-attention (MSA) and feed-forward neural networks (FFN), with layer normalization (LN) and residual connections typically used within each block:

$$\mathbf{x}_0 = [v_1, v_2, \cdots, v_P, \cdots, t_1, t_2, \cdots, t_N], \tag{1}$$

$$\mathbf{x}'_\ell = \mathrm{MSA}(\mathrm{LN}(\mathbf{x}_{\ell-1})) + \mathbf{x}_{\ell-1}, \ell \in \{1, \ldots, L\}, \tag{2}$$

$$\mathbf{x}_\ell = \mathrm{FFN}(\mathrm{LN}(\mathbf{x}'_\ell)) + \mathbf{x}'_\ell, \ell \in \{1, \ldots, L\}, \tag{3}$$

where $L$ is the layer number of LLM. The LVLM model generates an output text sequence $\mathcal{Y} = [y_1, y_2, \cdots, y_K] \in \mathbb{R}^{K \times D}$ by progressively generating each element, where $K$ represents the total length of the output text sequence. Then, the outputs are optimized through a generative loss in an auto-regressive manner (Liu et al., 2023c). The loss (the main loss) is formulated as:

$$\mathcal{L}_{\mathrm{main}}(\theta) = -\sum_{i=1}^{K} \log p\left(y_i \mid \mathcal{V}, \mathcal{T}, \mathcal{Y}_{<i}; \theta\right), \tag{4}$$

where $\mathcal{Y}_{<i}$ indicates output text tokens $[y_1, y_2, \cdots, y_{i-1}]$ ($i \geq 2$) and no output text tokens when $i{=}1$. $\theta$ indicates the trainable parameters. The main loss for the token $t_n$ is abbreviated as $\mathcal{L}_n(\theta)$.

**MoE**: The Mixture-of-Expert (MoE) layer is used to replace the FFN layer in this work, similar to (Lin et al., 2024; Dai et al., 2024). A MoE layer consists of multiple FFNs, each representing an expert, *i.e.*, $\mathcal{E} = [e_1, e_2, \cdots, e_E]$, where $E$ is the number of experts. For one token $t_n$, the router is typically a linear layer that predicts its probability of being assigned to each expert:

$$p_{\mathrm{moe}}(t_n)_i = \frac{e^{z_{\mathrm{moe}}(t_n)_i}}{\sum_{j=1}^{E} e^{z_{\mathrm{moe}}(t_n)_j}}, \tag{5}$$

where $z_{\mathrm{moe}}(t_n) = t_n \cdot \mathbf{W}$ and $p_{\mathrm{moe}}(t_n)_i$ is the routing score of $t_n$ for the $i$-th expert. The matrix $\mathbf{W} \in \mathbb{R}^{D \times E}$ represents the router parameters. We calculate a weighted sum of the outputs from the Top-$k$ experts with the highest softmax probabilities:

$$w_{\mathrm{moe}}(t_n)_i = \frac{e^{z_{\mathrm{moe}}(t_n)_i}}{\sum_{j=1}^{k} e^{z_{\mathrm{moe}}(t_n)_j}},$$

$$\mathrm{MoE}(t_n) = \sum_{i=1}^{k} w_{\mathrm{moe}}(t_n)_i \cdot e_i(t_n), \tag{6}$$

where $w_{\mathrm{moe}}(t_n)_i$ represents the weight of the $i$-th expert for $t_n$, and $e_i(t_n)$ is the output of the $i$-th expert. We express $\mathcal{L}_n(\theta)$ as $\mathcal{L}_n(\theta_{e_i}, \theta')$, where $\theta_{e_i}$ denotes the $i$-th expert, and $\theta'$ represents all parameters except for $\theta_{e_i}$. The visual token $v_n$ is the same as $t_n$ when passing the MoE.

**Our Method STGC**: In this work, we aim to use token-level gradients to model and solve data interference. First, tokens act as the basic unit during the forward pass of the LLM, so we can calculate the gradient generated from each token on the expert parameters, *i.e.*, token-level gradient. Then, as illustrated in Figure 2, our method consists of two steps: $(i)$ we compute token-level gradients and identify *conflicting tokens* by token-level gradients. During this process, we do not update any model parameters. $(ii)$ We add a regularization loss to the main loss to eliminate *conflicting tokens*. The details of these modules will be introduced in the subsequent sections.

## 3.2 CONFLICTING TOKEN IDENTIFICATION

Data interference in a LVLM is generated from interference between tokens within an expert. Some recent works model data interference through instruction embeddings (Chen et al., 2023d; Shen et al., 2024; Gou et al., 2023), *i.e.*, interference occurs when samples have distinct instruction embeddings. Alternatively, decisions can be made based on the specific task associated with each sample (Zhou et al., 2024). However, these works have two main limitations: $(i)$ features and labels jointly influence the parameter optimization direction, but these works rely on only one of these two factors. $(ii)$ These work operate at the sample level, whereas routing all tokens within a sample to the same expert does not solve interference between tokens (*e.g.*, visual and text tokens may be in interference) within the sample. To address the issues, we propose using token-level gradients, which can accurately depict the directions of parameter optimization at the token level, to identify interference between tokens within an expert.

First, we introduce the negative impact of the gradient conflict. Without loss of generality, we discuss two distinct instruction tokens, $t_n$ and $t_{n'}$, as shown in Figure 2 (a). Assume that both $t_n$ and $t_{n'}$ are processed by the expert $e_i$. Let $\mathbf{g}_n = \nabla_{\theta_{e_i}} \mathcal{L}_n(\theta_{e_i}, \theta')$ denote the gradient of the token $t_n$ with respect to the expert $\theta_{e_i}$. A small change in $\theta_{e_i}$ in the direction of $-\mathbf{g}_n$ is given by $\theta_{e_i} \leftarrow \theta_{e_i} - \delta\mathbf{g}_n$, with a step size $\delta$. The effect of this change on the loss of another token $t_{n'}$ is measured by $\Delta\mathcal{L}_{n'} = \mathcal{L}_{n'}(\theta_{e_i} - \delta\mathbf{g}_n, \theta') - \mathcal{L}_{n'}(\theta_{e_i}, \theta') = -\delta\mathbf{g}_n \cdot \mathbf{g}_{n'} + o(\delta)$, where the second equality is obtained by first-order Taylor approximation. Therefore, the model updating for $t_n$ is considered to negatively affect token $n'$ when $\mathbf{g}_n \cdot \mathbf{g}_{n'} < 0$, since it increases the loss of token $n'$, and vice versa. Thus, similar to (Yu et al., 2020), we define $\mathbf{g}_n$ and $\mathbf{g}_{n'}$ as conflicting gradients when their cosine similarity $\cos\phi_{nn'} < \tau$, where $\tau$ is a threshold and $\phi_{nn'}$ is the angle between $\mathbf{g}_n$ and $\mathbf{g}_{n'}$. Gradient conflicts cause the optimizer to converge to a sub-optimal solution.

We then define the *conflicting token*. Our goal is to adjust token routing to reduce gradient conflicts, but suddenly changing the routing of most tokens during training could lead to training instability. To stabilize training, we consider the expert as a whole to identify outliers as *conflicting tokens* for the expert. Let the tokens processed by the expert $e_i$ be denoted as $\{t_1, \cdots, t_{N_{e_i}}\}$, the *average gradient* on the expert $e_i$ is represented as:

$$\mathbf{g}_{mean} = \frac{\sum_{n=1}^{N_{e_i}} \mathbf{g}_n}{N_{e_i}}. \tag{7}$$

The *average gradient* indicates the holistic expert parameter updating direction at each iteration. When the gradient of a token and the *average gradient* are conflicting gradients, this token is detrimental to the learning of the expert $e_i$, so this token should be considered for assignment to another expert. Using the *average gradient* to identify *conflicting tokes* can keep most tokens in their current experts. A formal definition of a *conflicting token* is provided as follows:

**Definition 1 (Conflicting Token)** *The token $t_n$ is said to be a conflicting token if $\mathbf{g}_n$ and $\mathbf{g}_{mean}$ are conflicting gradients, where $\mathbf{g}_{mean}$ is the average gradient of all tokens in the expert of $t_n$.*

Lastly, we detail our method for identifying *conflicting tokens*, as illustrated in Figure 2 (a). Initially, we unfreeze only the expert layer in the MoE and compute the main loss. We then perform backpropagation to calculate the gradient produced by each token on the expert parameters (*i.e.*, token-level gradient). Subsequently, we calculate the *average gradient*, as well as the cosine similarity between the gradient of each token and the *average gradient*. Lastly, when the similarity is less than $\tau$, we mark the token as a *conflicting token*. During this process, we do not update any model parameters. Moreover, as the parameter size of each expert is very large, which leads to a very large parameter gradient size, we use an engineering trick, *i.e.*, using the gradients on part parameters as

an indicator, for reducing the GPU memory overhead to store gradients. Please refer to Sec. A of the supplementary material for more engineering implementation details. Identifying *conflicting token* allows us to design the regularization loss to reduce gradient conflicts in the next section.

## 3.3 CONFLICT ELIMINATION LOSS

The learning of a *conflicting token* increases the average loss of tokens within its current expert. Thus, after a *conflicting token* is identified, it should be reassigned to another expert for processing. To achieve this goal, we propose a simple yet effective regularization loss by constraining the routing scores predicted by the router, as shown in Figure 2 (b).

Specifically, we first identify the *conflicting tokens* for each expert within every MoE layer using token-level gradients. Then, the router predicts the routing logits $z_{\text{moe}}(t_n)$ of each *conflicting token* $t_n$. We record the current expert ID $id_{\text{moe},n}$ of each *conflicting token* $t_n$. For a LVLM, we use the recorded expert ID $id_{\text{moe},n}$ to calculate the loss:

$$
\begin{aligned}
z'_{\text{moe}}(t_n) &= -z_{\text{moe}}(t_n), \\
p'_{\text{moe}}(t_n)_i &= \frac{e^{z'_{\text{moe}}(t_n)_i}}{\sum_{j=1}^{E} e^{z'_{\text{moe}}(t_n)_j}}, \\
\mathcal{L}_{\text{CEL}} &= \frac{1}{N_{all} \cdot E} \sum_{n=1}^{N_{all}} \sum_{i=1}^{E} \log(p'_{\text{moe}}(t_n)_i) \cdot q_{\text{moe}}(t_n)_i,
\end{aligned}
\tag{8}
$$

where $N_{all}$ is the count of all *conflicting tokens*, $E$ is the number of experts, and $p'_{\text{moe}}(t_n)$ represents the inverted routing score for the *conflicting token* $t_n$. The $q_{\text{moe}}(t_n)$ define one-hot vectors, with $q_{\text{moe}}(t_n)_{id_{\text{moe},n}} = 1$. This loss is designed to encourage the reassignment of *conflicting tokens* from their current experts to other experts.

## 3.4 TOTAL LOSS

To encourage experts to handle tokens in a balanced manner, the differentiable load balancing loss, as introduced in (Fedus et al., 2022), is typically defined for each MoE layer as follows:

$$
\mathcal{L}_{\text{aux}} = E \cdot \sum_{i=1}^{E} \mathcal{F}_i \cdot \mathcal{P}_i,
\tag{9}
$$

where $\mathcal{F}_i$ indicates the fraction of tokens processed by each expert $e_i$, and $\mathcal{P}_i$ indicates the average routing score of tokens assigned to each expert $e_i$. We also use $\mathcal{L}_{\text{aux}}(\theta)$ to denote the average load balance loss of all MoE layers for convenience.

In conclusion, the total loss to update the model parameters $\theta$ is defined as:

$$
\mathcal{L}_{\text{total}} = \mathcal{L}_{\text{moe}} + \beta \cdot \mathcal{L}_{\text{CEL}} = (\mathcal{L}_{\text{main}} + \alpha \cdot \mathcal{L}_{\text{aux}}) + \beta \cdot \mathcal{L}_{\text{CEL}},
\tag{10}
$$

where $\mathcal{L}_{\text{moe}}$ indicates the loss used in existing MoE-based LVLMs. $\alpha$ and $\beta$ are hyper-parameters. We use $\mathcal{L}_{\text{main}}$ to denote $\mathcal{L}_{\text{main}}(\theta)$ for convenience, and the same applies to other losses.

# 4 EXPERIMENTS

## 4.1 EXPERIMENTAL SETUP

**Benchmark**: Some academic-task-oriented and instruction-following benchmarks are collected for evaluating the LVLM. For academic-task-oriented benchmarks, VQA-v2 (Goyal et al., 2017b) and GQA (Hudson & Manning, 2019) assess the visual perception capabilities of models through open-ended short answers. VizWiz (Gurari et al., 2018) evaluates the zero-shot generalization of models on visual questions asked by visually impaired people. ScienceQA (Lu et al., 2022), a multiple-choice benchmark, evaluates the zero-shot generalization of models on scientific question answering. TextVQA (Singh et al., 2019a) focuses on text-rich visual question answering tasks. ChartQA (Masry et al., 2022) focuses on visual and logical reasoning tasks over charts. DocVQA (Mathew et al., 2021) focuses on reading comprehension tasks over document images.

Table 1: **Comparison between different LVLMs on image understanding benchmarks.** "Act.", "V", "Q", "P", "M", and "S" represent activated parameters, Vicuna (Chiang et al., 2023), Qwen (Bai et al., 2023a), Phi-2 (Microsoft, 2023), MobileLLaMA (Chu et al., 2023), and StableLM (Team), respectively. Main evaluation Benchmarks include VQA$^{v2}$ (Goyal et al., 2017a); GQA (Hudson & Manning, 2019); VisWiz (Gurari et al., 2018); SQA$^I$: ScienceQA-IMG (Lu et al., 2022); VQA$^T$: TextVQA (Singh et al., 2019b); POPE (Li et al., 2023a); MME (Fu et al., 2023); MMB: MMBench (Liu et al., 2023d); MM-Vet (Yu et al., 2023a). $*$ indicates that there is some overlap in the training data. $\dagger$ denotes the use of a stronger visual encoder (siglip-so400m-patch14-384). All "Sparse Model" methods use the configure 4Top2. We calculate the average performance across all datasets except for MME, naming it "Avg". In the table below, we report the best performance that we achieve when activating only 3.6B parameters during inference.

| Method | LLM | Act. | VQA$^{v2}$ | GQA | VisWiz | SQA$^I$ | VQA$^T$ | POPE | MME | MMB | MM-Vet | AI2D | ChartQA | DocVQA | Avg |
|---|---|---|---|---|---|---|---|---|---|---|---|---|---|---|---|
| *Dense Model* | | | | | | | | | | | | | | | |
| LLaVA-1.5 | V-13B | 13B | 80.0* | 63.3* | 53.6 | 71.6 | 61.3 | 85.9 | 1531.3 | 67.7 | 35.4 | 49.6 | 18.1 | 24.0 | 55.5 |
| Qwen-VL | Q-7B | 6.7B | 78.8* | 59.3* | 35.2 | 67.1 | 63.8 | - | - | 38.2 | - | - | - | - | - |
| LLaVA-1.5 | V-7B | 6.7B | 78.5* | 62.0* | 50.0 | 66.8 | 58.2 | 85.9 | 1510.7 | 63.4 | 30.5 | - | - | - | - |
| TinyGPT-V | P-2.7B | 2.7B | - | 33.6* | 33.4 | - | - | - | - | - | - | - | - | - | - |
| MobileVLM | M-2.7B | 2.7B | - | 59.0* | - | 61.0 | 47.5 | 84.9 | 1288.9 | 59.6 | - | - | - | - | - |
| LLaVA-Phi | P-2.7B | 2.7B | 71.4* | - | 35.9 | 68.4 | 48.6 | 85.0 | 1335.1 | 59.8 | 28.9 | - | - | - | - |
| *Sparse Model* | | | | | | | | | | | | | | | |
| MoE-LLaVA | S-1.6B | 2.0B | 76.7* | 60.3* | 36.2 | 62.6 | 50.1 | 85.7 | 1318.2 | 60.2 | 26.9 | 48.8 | 15.3 | 18.4 | 49.2 |
| MoE-LLaVA | P-2.7B | 3.6B | 77.6* | 61.4* | 43.9 | 68.5 | 51.4 | 86.3 | 1423.0 | 65.2 | 34.3 | 58.8 | 19.9 | 21.5 | 53.5 |
| DYNMOE-LLaVA | P-2.7B | 3.4B | 77.9* | 61.6* | 45.1 | 68.0 | 51.8 | 86.0 | 1429.6 | 66.6 | 33.6 | - | - | - | - |
| MoE-LLaVA$^\dagger$ | P-2.7B | 3.6B | 79.9* | 62.6* | 43.7 | 70.3 | 57.0 | 85.7 | 1431.3 | 68.0 | 35.9 | 59.5 | 15.4 | 25.6 | 54.9 |
| Our Method$^\dagger$ | P-2.7B | 3.6B | 80.0* | 63.0* | 48.6 | 70.9 | 58.8 | 86.5 | 1481.7 | 71.0 | 40.7 | 64.5 | 44.7 | 42.1 | 61.0 |

For instruction-following benchmarks, POPE (Li et al., 2023b) evaluates the degree of hallucination in model responses on three sampled subsets of COCO (Lin et al., 2014): Random, Common, and Adversarial. MME (Fu et al., 2023) assesses the visual perception of models with yes/no questions. MMBench (Liu et al., 2023d) evaluates the robustness of model answers with all-round shuffling on multiple choice answers. MM-Vet (Yu et al., 2023b) evaluates the model capabilities in engaging in visual conversations on a diverse range of tasks and evaluates the correctness and helpfulness of the responses using the GPT-4 evaluation framework. AI2D (Kembhavi et al., 2016), a multiple-choice benchmark, evaluates the model capabilities for science diagram comprehension.

**Baseline**: Our main baseline is MoE-LLaVA (Lin et al., 2024). MoE-LLaVA incorporates a MoE into LVLMs and proposes a three-stage training scheme. It trains only the MoE in the third stage, *i.e.*, the instruction tuning stage. MoE-LLaVA has four experts and selects the Top-2 experts to handle tokens, and we refer to this configuration as 4Top2. Building on MoE-LLaVA, we add a novel regularization loss $\mathcal{L}_{\text{CEL}}$ during the instruction tuning stage to enhance the MoE. For the language model backbone, we follow MoE-LLaVA to use StableLM-1.6B and Phi2-2.7B. The visual encoder is usually set as clip-vit-large-patch14-336. $\alpha$=0.01, following MoE-LLaVA. We also compare with DYNMOE-LLaVA (Guo et al., 2024), which improves the MoE-LLaVA by dynamically setting the expert count. For more implementation details, please refer to Sec. A of supplementary material.

## 4.2 IMAGE UNDERSTANDING EVALUATION

**Image Question Answering**: We evaluate the performance of our method on five image question-answering benchmarks, as shown in Table 1, and report the number of activated parameters as a measure of efficiency. Our method demonstrates superior image understanding capabilities, achieving the 80.0%, 63.0%, 48.6%, 70.9%, and 58.8% performance on VQA$^{v2}$, GQA, VisWiz, SQA$^I$, and VQA$^T$, respectively. Compared to LLaVA-1.5 with 7B activated parameters, our method brings 1.5%, 1.0%, 4.1%, and 0.6% performance increase on VQA$^{v2}$, GQA, SQA$^I$, and VQA$^T$, respectively, when only activating 3.6B parameters.

**Benchmark Toolkit**: To comprehensively evaluate the multi-modal understanding capabilities of our method, we assess its performance across four benchmark toolkits in Table 1. These toolkits typically serve as tools to verify the model ability to engage in natural language questioning. As shown in Table 1, our method achieves 86.5%, 1481.7, 71.0%, and 40.7% performance on POPE,

Table 2: **STGC as a plug-in**. We set different baselines and add the proposed STGC. 4Top1 means four experts are set, and the Top-1 expert is selected to handle tokens. $^\dagger$ indicates that a stronger visual encoder (siglip-so400m-patch14-384) is used. We calculate the average performance across all datasets except for MME, naming it "Avg".

| Method | LLM | Act. | VQA$^{v2}$ | GQA | VisWiz | SQA$^I$ | VQA$^T$ | POPE | MME | MMB | MM-Vet | Avg |
|---|---|---|---|---|---|---|---|---|---|---|---|---|
| MoE-LLaVA-4Top1 | S-1.6B | 1.6B | 74.5* | 58.6* | 25.7 | 55.8 | 45.0 | 85.2 | 1245.3 | 56.2 | 27.2 | 53.5 |
| +STGC | S-1.6B | 1.6B | 74.9* | 59.4* | 27.4 | 57.5 | 46.5 | 85.8 | 1276.8 | 56.8 | 28.5 | 54.6 |
| MoE-LLaVA-4Top2 | S-1.6B | 2.0B | 76.7* | 60.3* | 36.2 | 62.6 | 50.1 | 85.7 | 1318.2 | 60.2 | 26.9 | 57.3 |
| +STGC | S-1.6B | 2.0B | 76.9* | 60.9* | 37.7 | 62.6 | 50.7 | 85.9 | 1355.1 | 60.7 | 28.2 | 58.0 |
| MoE-LLaVA-4Top2 | P-2.7B | 3.6B | 77.6* | 61.4* | 43.9 | 68.5 | 51.4 | 86.3 | 1423.0 | 65.2 | 34.3 | 61.1 |
| +STGC | P-2.7B | 3.6B | 78.0* | 62.1* | 47.2 | 68.1 | 52.3 | 86.9 | 1429.2 | 66.7 | 33.3 | 61.8 |
| MoE-LLaVA-4Top2$^\dagger$ | P-2.7B | 3.6B | 79.9* | 62.6* | 43.7 | 70.3 | 57.0 | 85.7 | 1431.3 | 68.0 | 35.9 | 62.9 |
| +STGC | P-2.7B | 3.6B | 80.3* | 63.2* | 45.1 | 70.3 | 57.4 | 86.1 | 1447.6 | 69.7 | 35.7 | 63.5 |

Table 3: **Scalability of STGC**. We consider employing the larger scale of data. With more data, the STGC can bring a more significant performance increase.

| Method | LLM | Data | VQA$^{v2}$ | GQA | VisWiz | SQA$^I$ | VQA$^T$ | POPE | MME | MMB | MM-Vet | Avg |
|---|---|---|---|---|---|---|---|---|---|---|---|---|
| MoE-LLaVA-4Top2$^\dagger$ | P-2.7B | 665K | 79.9* | 62.6* | 43.7 | 70.3 | 57.0 | 85.7 | 1431.3 | 68.0 | 35.9 | 62.9 |
| +STGC | P-2.7B | 665K | 80.3* | 63.2* | 45.1 | 70.3 | 57.4 | 86.1 | 1447.6 | 69.7 | 35.7 | 63.5 |
| MoE-LLaVA-4Top2$^\dagger$ | P-2.7B | 1021K | 79.7* | 63.0* | 42.7 | 71.1 | 56.9 | 84.3 | 1439.9 | 70.4 | 42.2 | 63.8 |
| +STGC | P-2.7B | 1021K | 80.0* | 63.0* | 48.6 | 70.9 | 58.8 | 86.5 | 1481.7 | 71.0 | 40.7 | 64.9 |

Table 4: **Study about routing strategies.** *embedding-based*, *feature-based*, and *task-based* correspond to routing strategies similar to MoCLE (Gou et al., 2023), LoRA-MoE (Chen et al., 2023d), and MoCLE (Zhou et al., 2024), respectively. The configure MoE-LLaVA-4Top2 with StableLM-1.6B is set as the baseline.

| Method | VQA$^{v2}$ | GQA | VisWiz | SQA$^I$ | VQA$^T$ | POPE | MME | MMB | MM-Vet | Avg |
|---|---|---|---|---|---|---|---|---|---|---|
| MoE-LLaVA | 76.7* | 60.3* | 36.2 | 62.6 | 50.1 | 85.7 | 1318.2 | 60.2 | 26.9 | 57.3 |
| +*embedding-based* | 75.8* | 57.0* | 34.0 | **63.7** | 50.3 | **86.1** | 1312.8 | 61.3 | 27.3 | 56.9 |
| +*feature-based* | 75.7* | 58.1* | 36.9 | 63.2 | 50.0 | 85.9 | 1338.8 | **61.5** | 26.6 | 57.2 |
| +*task-based* | 73.6* | 58.2* | 29.0 | **63.7** | 49.2 | 81.5 | 1306.3 | 59.5 | 25.2 | 48.9 |
| +STGC | **76.9*** | **60.9*** | **37.7** | 62.6 | **50.7** | 85.9 | **1355.1** | 60.7 | **28.2** | **58.0** |

MME, MMB, and MM-Vet, respectively. Compared to LLaVA-1.5 with 7B activated parameters, our method brings 0.6%, 7.6%, and 10.2% performance increase on POPE, MMB, and MM-Vet, respectively, when only activating 3.6B parameters.

## 4.3 ABLATION STUDY

**STGC as a plug-in**: We design the STGC as a plug-in for existing MoE methods. To verify the robustness and effectiveness of the proposed STGC as a plug-in, we select different baselines, including different MoE configures (4Top2 & 4Top1), LLM (S-1.6B & P-2.7b), and visual encoders (clip & siglip). Then, we add the STGC onto these baselines. As shown in Table 2, adding STGC always brings a stable and convincing performance increase under diverse baselines.

**Scalability of STGC**: One necessary hypothesis of this work is that there is severe interference between diverse training data, so STGC should bring a more significant performance increase when using a more complex dataset to train. To verify this, we expand the size of the public dataset we are using for further experiments. Specifically, we use the training data provided in the Open-LLaVA-NeXT project (Lin & Long, 2024). As shown in Table 3, under more data, STGC brings a more significant average performance increase (1.1% *vs.* 0.6%).

Table 5: **Study about hyper-parameter sensitivity.** Settings for results in Table 2 are highlighted in blue . The configure MoE-LLaVA-4Top2 with StableLM-1.6B is set as the baseline.

(a) **The threshold for identifying *conflicting tokens*.**

| $\tau$ | GQA | VisWiz | VQA$^{\text{T}}$ | MMB | MM-Vet |
|---|---|---|---|---|---|
| 0.1 | 60.6 | 35.1 | **50.9** | 61.3 | 25.6 |
| 0.0 | **60.9** | **37.7** | 50.7 | 60.7 | **28.2** |
| -0.1 | 60.6 | 34.9 | 50.5 | **61.4** | 25.9 |

(b) **The weighting of our proposed loss.**

| $\beta$ | GQA | VisWiz | VQA$^{\text{T}}$ | MMB | MM-Vet |
|---|---|---|---|---|---|
| 0.5 | 60.5 | 35.6 | 50.5 | **61.4** | 26.9 |
| 1.0 | **60.9** | **37.7** | 50.7 | 60.7 | **28.2** |
| 2.0 | 60.6 | 35.9 | **50.9** | 60.6 | 27.2 |

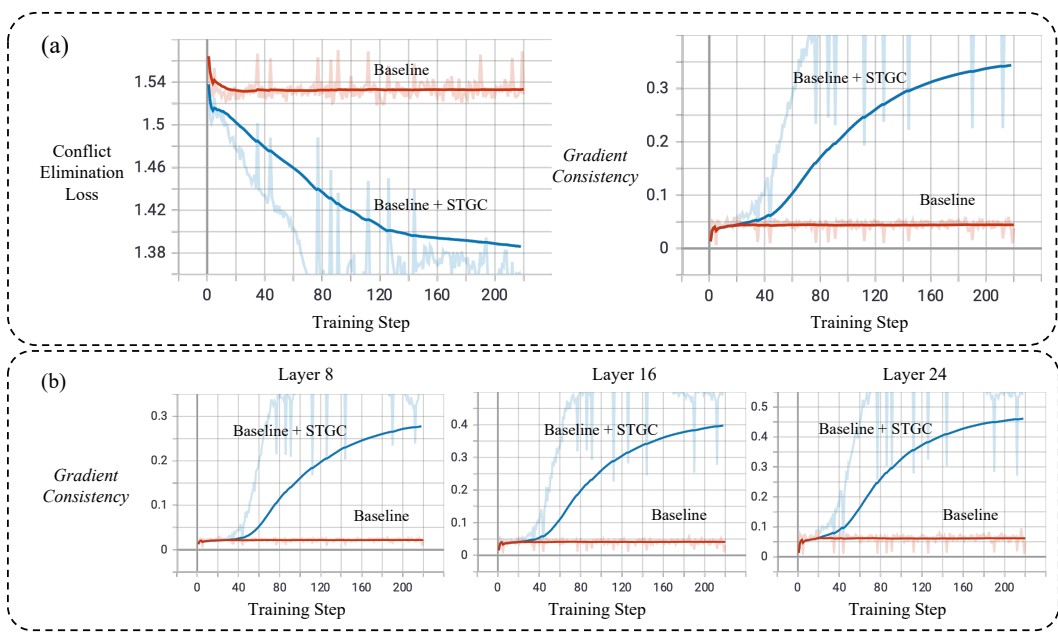

Figure 3: **Statistical verification**. We conduct a deep analysis of the role of STGC. "Baseline" indicates MoE-LLaVA. "Baseline + STGC" indicates our method. (a) We compute a novel metric, *gradient consistency* (the mean cosine similarity between gradients of all tokens within an expert), for verifying that the decrease of the proposed loss leads to the more consistent token gradients within an expert. (b) We further analyze the *gradient consistency* on different layers.

**Study about Different Routing Strategies**: We compare with different routing strategies in Table 4. Some works (Chen et al., 2023d; Gou et al., 2023; Zhou et al., 2024) propose using instruction embeddings (features) or sample task labels to design routing strategies. Since they are designed for LoRA-MoE, we directly replace the FFN with the MoE structure, making it difficult to compare the performance of STGC with that in their papers. To this end, we re-implemented these methods in MoE-LLaVA. Specifically, ($i$) *embedding-based*. Similar to MoCLE (Gou et al., 2023), we use the K-means algorithm to cluster instruction embeddings into 128 groups before training and conduct the routing based on the cluster results. ($ii$) *feature-based*. Similar to LoRA-MoE (Chen et al., 2023d), we use instance-level instruction token average representation to predict routing scores. ($ii$) *task-based*. Similar to (Zhou et al., 2024), we categorize samples into four experts based on their tasks. For more details, please refer to Sec. B of supplementary material. As shown in Table 4, the performance does not increase on most datasets when using sample-level embedding or task information. Using token-level gradients achieves a significantly higher average performance improvement.

**Sensitivity Study of Different Hyper-parameters**: We conduct the sensitivity study of STGC in Table 5 for two main hyper-parameters, the threshold $\tau$ of identifying *conflicting tokens* and the loss weighting $\beta$. First, when the gradient $\mathbf{g}_n$ of the token $t_n$ and and the *average gradient* $\mathbf{g}_{mean}$ satisfy the condition $\cos \phi_{nmean} < \tau$, we flag the token as a *conflicting token*. We discuss different

thresholds of identifying *conflicting tokens*: $\tau \in \{0.1, 0.0, -0.1\}$. Second, we discuss different loss weightings $\beta \in \{0.5, 1.0, 2.0\}$. As shown in Table 5, we find: ($i$) When $\tau = 0$, the performance on most datasets is the best. The results are consistent with the common belief, *i.e.*, gradients are considered conflicting when their cosine similarity is less than 0. ($ii$) The proposed loss is relatively robust to different loss weightings $\beta$, with the highest performance on most datasets when $\beta$=1.0.

**Statistical Verification**: One main claim of this work is that STGC can effectively reduce the gradient conflicts between tokens within an expert, *i.e.*, making the gradient directions of tokens within an expert more consistent, thus decreasing data interference and improving model performance. To verify this, we design a statistical verification experiment. In this experiment, the training consists of two steps: The first step is to gather gradients of all tokens within an expert $e_i$ and compute their cosine similarity to form a similarity matrix. Then, we define the mean of the similarity matrix as $sim_i$. We define the average of $sim_i$ on all experts as *gradient consistency*, serving as a novel metric to evaluate whether the gradient directions are consistent within the expert. The second step is to update models. In this experiment, we only use the proposed loss to update parameters, for undisturbedly observing its impact on the *gradient consistency*. In experiments, we fed one sample into the LVLM per device for each forward pass, and the gradient accumulation step is 16.

We first present the curve graph of the proposed loss and the *gradient consistency* obtained from the TensorBoard dashboard in Figure 3 (a). Then, we present the *gradient consistency* on different layers in Figure 3 (b); The total layer number is 32, and we analyze the layers $\{8, 16, 24\}$. We find: ($i$) By decreasing the proposed conflict elimination loss, the *gradient consistency* significantly improves, indicating that the token gradient directions within an expert become more consistent. This verifies the role of STGC in reducing gradient conflicts. ($ii$) For different layers, STGC always increases the *gradient consistency*. The deeper layers seem to have higher *gradient consistency* after adding STGC, while the shallower layers have lower *gradient consistency*.

## 5 CONCLUSION AND LIMITATIONS

Our study reveals that there is the interference between tokens within an expert for the MoE, leading to sub-optimal learning for the expert. To reduce the interference between tokens, we propose employing token-level gradients to identify *conflicting tokens*, and then adding a novel conflict elimination loss to optimize token routing based on *conflicting tokens*. Our method STGC acts as a plug-in, which can be easily integrated into existing MoE-based LVLMs. Extensive experiments demonstrate the effectiveness of STGC across diverse datasets. Especially when the data diversity is larger, our method brings a more significant performance increase.

Limitation. Since in each iteration, a token only passes one expert and not the others in each MoE layer, its gradient only reflects whether it conflicts with the holistic optimization direction of its current expert, but it is difficult to define its relationship with other experts. Thus, although it has been confirmed that the proposed solution can solve token gradient conflicts by optimizing token routing, the solution still has room for improvement, as it is challenging to determine the optimal expert of a token. In the future, exploring whether it is possible to determine the optimal expert of a token from the optimization perspective may be an intriguing direction.

**Acknowledgements.** This work is supported in part by National Science Foundation for Distinguished Young Scholars under Grant 62225605, Zhejiang Provincial Natural Science Foundation of China under Grant LD24F020016, "Pioneer" and "Leading Goose" R&D Program of Zhejiang (No. 2024C01020), Project 12326608 supported by NSFC, the Ningbo Science and Technology Innovation Project (No.2024Z294), and is supported by Kuaishou Technology.

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

## 6 SUPPLEMENTARY MATERIAL

In our supplementary material, we provide the following details and experiments:

- section 6.1: We provide more engineering implementation details about training.
- section 6.2: We provide more implementation details about sample-level routing.
- section 6.3: We provide more experimental results about expert loading, loss design, token gradient statistic, and different routing mechanisms.
- section 6.4: We provide a analysis about computational overhead.
- section 6.5: We provide experimental results about language tasks.
- section 6.6: We provide a brief theoretical analysis.
- section 6.7: We provide a deeper analysis about gradient conflicting phenomenon.

### 6.1 IMPLEMENTATION DETAILS ABOUT TRAINING

**Token-level gradient**: Each expert is an FFN containing multiple linear layers. For instance, the FFN in Phi-2 (Microsoft, 2023) includes two linear layers, $fc^1$ and $fc^2$. Assuming there are $E \times L$ experts, $[e_1, e_2, \cdots, e_{E \times L}]$, where $E$ is the number of experts in each MoE and $L$ is the MoE layer number of LLM. In expert $e_i$, the weights corresponding to the two linear layers are $w_i^1 \in \mathbb{R}^{D \times D'}$ and $w_i^2 \in \mathbb{R}^{D' \times D}$, and the biases are $b_i^1 \in \mathbb{R}^{D'}$ and $b_i^2 \in \mathbb{R}^D$ respectively, where $D$ is the hidden size of LLM and $D'$ is the intermediate size.

First, during the forward pass of a batch, we freeze the parameters except for the biases and compute the main loss. Then, we perform a backward pass, using the Operator "call_for_per_sample_grads" provided by PyTorch to capture the gradients $\mathbf{g}_n^1 \in \mathbb{R}^D$ and $\mathbf{g}_n^2 \in \mathbb{R}^{D'}$ of the token $t_n$ on the biases. Next, we calculate the *average gradients* $\mathbf{g}_{mean}^1$ and $\mathbf{g}_{mean}^2$. Let the tokens processed by the expert $e_i$ be denoted as $\{t_1, \cdots, t_{N_{e_i}}\}$, the *average gradients* are represented as:

$$
\begin{aligned}
\mathbf{g}_{mean}^1 &= \frac{\sum_{n=1}^{N_{e_i}} \mathbf{g}_n^1}{N_{e_i}}, \\
\mathbf{g}_{mean}^2 &= \frac{\sum_{n=1}^{N_{e_i}} \mathbf{g}_n^2}{N_{e_i}}
\end{aligned}
\tag{11}
$$

where $\mathbf{g}_{mean}^1 \in \mathbb{R}^D$ and $\mathbf{g}_{mean}^2 \in \mathbb{R}^{D'}$. Following that, we compute the cosine similarity between $\mathbf{g}_n^1$ and $\mathbf{g}_{mean}^1$ as $s_n^1$ and the cosine similarity between $\mathbf{g}_n^2$ and $\mathbf{g}_{mean}^2$ as $s_n^2$. Thus, for the token $t_n$, the similarity metric $s_n$ is defined as:

$$
s_n = \frac{s_n^1 + s_n^2}{2}
\tag{12}
$$

Finally, we identify the *conflicting token*: *when $s_n$ is lower than the threshold $\tau$, the token $t_n$ is a conflicting token.*

*Why to use the gradients on the biases?* This engineering trick brings two advantages. $(i)$ Assume the gradients of the token $t_n$ on the weights be $\mathbf{g}_n^{1,w}$ and $\mathbf{g}_n^{2,w}$ respectively. $\mathbf{g}_n^{1,w} \in \mathbb{R}^{D \times D'}$ and $\mathbf{g}_n^{2,w} \in \mathbb{R}^{D' \times D}$. Thus, we need a significant GPU memory overhead to store gradients. If storing only the gradients on the biases, the GPU memory overhead significantly reduces. $(ii)$ When the parameter size in the computational graph is larger, the backward pass is longer. We compute only the gradients on the biases, so it is very fast to capture gradients. *Whether does this operation work?* Similar to the computation of $s_n$, we use $\mathbf{g}_n^{1,w}$ and $\mathbf{g}_n^{2,w}$ to compute the similarity metric $s_n^w$ of the token $t_n$. We then compute the Pearson correlation coefficient between $s_n^w$ and $s_n$ ($n \in \{1, \cdots\}$) and find the value is usually larger than 0.9. Thus, we believe that $s_n$ reveals the relationship between the token gradient and the *average gradient* in the expert well. Our experiments also verify that the proposed STGC can increase performance.

After identifying *conflict tokens*, we add the parameters that need to be updated into the optimizer (we follow the training configure of MoE-LLaVA). We add the conflict elimination loss to optimize the router based on the identification of *conflict tokens*.

Table 6: **Hyper-parameters in training.**

| | Epoch | Learning rate | Learning rate schedule | Weight decay |
|---|---|---|---|---|
| Instruction Tuning | 1 | 2e-5 | Cosine | 0.0 |

| | Text max length | Batch size per GPU | GPU | Precision |
|---|---|---|---|---|
| Instruction Tuning | 2048 | 16 | 8×A800-80G | Bf16 |

| Data | Size | Response formatting prompts |
|---|---|---|
| LLaVA | 158K | – |
| ShareGPT | 40K | – |
| VQAv2 | 83K | Answer the question using a single word or phrase. |
| GQA | 72K | |
| OKVQA | 9K | |
| OCRVQA | 80K | |
| A-OKVQA | 66K | Answer with the option's letter from the given choices directly. |
| TextCaps | 22K | Provide a one-sentence caption for the provided image. |
| RefCOCO | 48K | *Note: randomly choose between the two formats* Provide a short description for this region. |
| VG | 86K | Provide the bounding box coordinate of the region this sentence describes. |
| Total | 665K | |

Table 7: Instruction-following data mixture. The data is from LLaVA-1.5 (Liu et al., 2023b).

**Training Scheme**: Our training scheme follows MoE-LLaVA (Lin et al., 2024). The details are presented in Table 6. During instruction fine-tuning, we use a batch size of 128 and a learning rate of 2e-5. We directly use the pre-trained models from MoE-LLaVA (Lin et al., 2024) to conduct instruction tuning.

**Training Datasets**: We use LLaVA-mix-665k (Liu et al., 2023b) as instruction tuning training data to conduct most experiments. The data structure is presented in Table 7. To verify the scalability of the STGC model, we conducted experiments using 1021K data from the Open-LLaVA-NeXT dataset (Lin & Long, 2024). In Table 1 of the main paper, we report the best performance that we achieve when activating only 3.6B parameters during inference, using 1021K data for training.

### 6.2 IMPLEMENTATION DETAILS ABOUT SAMPLE-LEVEL ROUTING SCHEMES

*Embedding-based*: Similar to MoCLE (Gou et al., 2023), we encode all the instructions of different datasets using the all-MiniLM-L6-v2 variant of the Sentence Transformer model (Reimers, 2019) and cluster their embeddings via K-means clustering algorithm. After clustering, following the practice of MoCLE, we initialize K learnable embeddings, and each embedding corresponds to a cluster center. When a sample belongs to the $k$-th cluster center, the $k$-th learnable embedding is extracted and fed into the router to predict routing scores. In our experiments, we set K=128. Following the practice of MoCLE, we do not add the load balance loss.

*Feature-based*: Similar to LoRA-MoE (Chen et al., 2023d), we take the average of instruction token representations of each instance as input to predict its routing scores in each expert. Then, the Top-$k$ experts are selected based on routing scores for each sample to generate the prediction. Following the practice of LoRA-MoE, we do not add the load balance loss.

*Task-based*: Similar to MoLA (Zhou et al., 2024), we promote similarity in routing between data from the same task while emphasizing distinctiveness in routing between data from different tasks. We employ LLaVA-mix-665k (Liu et al., 2023b) to conduct experiments, significantly different from the used data in MoLA (Zhou et al., 2024). Therefore, we empirically divide the data into four types of tasks.: ($i$) *Caption*. For example, the instruction is "Provide a one-sentence caption for the provided image.". ($ii$) *VQA*. For example, the instruction is "Answer the question using a single

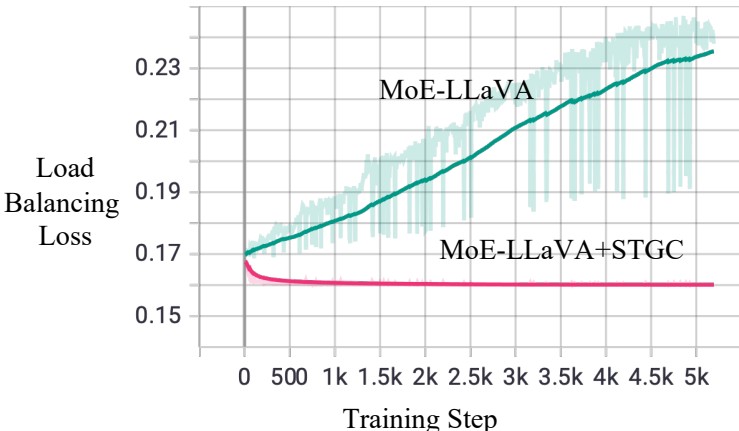

Figure 4: **Load balance loss**. "Baseline" indicates MoE-LLaVA. "Baseline+STGC" indicates our method. We present the load balancing loss curve before and after adding STGC. The results are obtained from the regular training. The total training step count is 5194 for an epoch. When the load balancing loss is lower, the expert load is more balanced.

Table 8: **Study about the load balance loss weighting** $\alpha$. The configure MoE-LLaVA-4Top2 with StableLM-1.6B is set for experiments.

| | $\alpha$ | VQA$^{v2}$ | GQA | VisWiz | SQA$^I$ | VQA$^T$ | POPE | MME | MMB | MM-Vet | Avg |
|---|---|---|---|---|---|---|---|---|---|---|---|
| MoE-LLaVA | 0.01 | 76.7* | 60.3* | 36.2 | 62.6 | 50.1 | 85.7 | 1318.2 | 60.2 | 26.9 | 57.3 |
| MoE-LLaVA | 0.1 | 75.7* | 59.7* | 37.7 | 61.3 | 49.9 | 85.6 | 1338.0 | 60.4 | 27.2 | 57.2 |

word or phrase.". $(iii)$ *OCR*, including all data in OCRVQA. $(iv)$ *Region-aware*. For example, the instruction is "Provide a short description for this region.". The expert label of *Caption*, *VQA*, *OCR*, or *Region-aware* data is 0, 1, 2, or 3, respectively. The ratio of *Caption*, *VQA*, *OCR*, or *Region-aware* data is 3.5%, 61.6%, 12.8%, and 22.1%, respectively.

## 6.3 MORE EXPERIMENTAL RESULTS

### 6.3.1 EXPERT LOADING

**Loss Curve**: We present the the load balancing loss curve in Figure 4. As shown in Figure 4, the proposed STGC benefits the decrease of the load balancing loss. A possible reason is that the expert load imbalance means that many tokens are routed to an expert, significantly increasing the possibility that tokens have gradient conflicts. After adding the STGC, some tokens are moved from the "crowded" expert (many tokens) to the "empty" expert (few tokens). This may be also a new perspective on why the load balance is important to the MoE system.

**Additional Ablations on** $\alpha$: $\alpha$ is the weighting of the load balance loss. We discuss different loss weightings $\alpha \in \{0.01, 0.1\}$ (0.01 is the standard value set in MoE-LLaVA (Lin et al., 2024)). As shown in Table 8, we find that increasing the weighting of the load balance loss degenerates the model performance.

**Visualization of Expert Loading**: we follow MoE-LLaVA (Lin et al., 2024) to obtain the distribution of expert loading and the visualization of the activated pathways. The distribution of expert loading examines the expert use frequency for all tokens (Lin et al., 2024). Activated pathways examine the behavior of experts at the token level (Lin et al., 2024): this visualization tool tracts the activated pathways of all tokens on validation datasets; given all activated pathways, the visualization tool employs PCA to obtain the top-10 pathways. As shown in Figure 5, we find that $(i)$ STGC benefits the expert load balance. A possible reason is that the expert load imbalance means that many tokens are routed to an expert, significantly increasing the possibility that tokens have gradient conflicts. After adding the STGC, some tokens are moved from the "crowded" expert (many

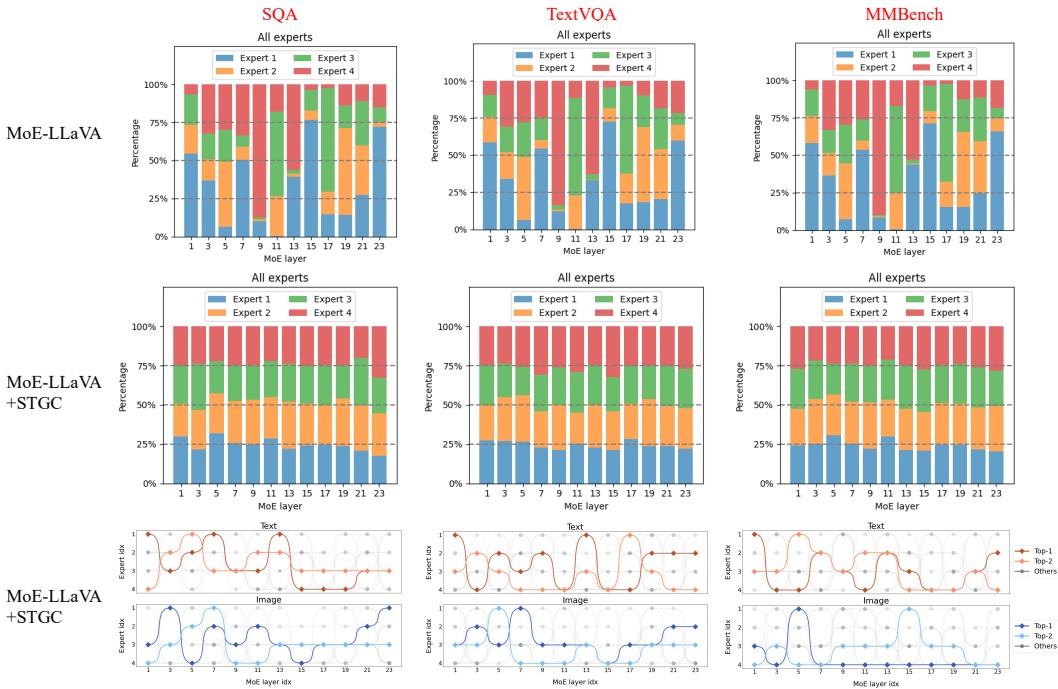

Figure 5: **Expert Loading and activated pathways**. The configure MoE-LLaVA-4Top2 with StableLM-1.6B is set for experiments We select three validation datasets, *i.e.*, SQA (Lu et al., 2022), TextVQA (Singh et al., 2019a), and MMBench (Liu et al., 2023d), to analyze expert loading and activated pathways. In activated pathways, the colorful paths represent the top-2 paths for text and image, respectively, while the gray paths represent the remaining 8 paths.

tokens) to the "empty" expert (few tokens). This further validates that STGC would effectively utilize each expert, instead of collapsing into only using one expert, which is also a new perspective on why the load balance is important to the MoE system. $(ii)$ The activated pathways are significantly different for SQA (Lu et al., 2022), TextVQA (Singh et al., 2019a), and MMBench (Liu et al., 2023d). This implies that although the distribution of expert load across different datasets is similar, the token routing behavior is still significantly different among datasets, *i.e.*, different tokens have been assigned to various experts.

### 6.3.2 LOSS DESIGN

The goal of the conflict elimination loss is to reduce the routing score $p_{\mathrm{moe}}(t_n)$ of the *conflicting token* $t_n$ on its current expert. We discuss different designs for the conflict elimination loss: $(i)$ MSE-like: Simply setting the routing score $p_{\mathrm{moe}}(t_n)$ to the minimum.

$$p_{\mathrm{moe}}(t_n)_i = \frac{e^{z_{\mathrm{moe}}(t_n)_i}}{\sum_{j=1}^{E} e^{z_{\mathrm{moe}}(t_n)_j}},$$

$$\mathcal{L}_{\mathrm{CEL}}^{\mathrm{MSE}} = \frac{1}{N_{all} \cdot E} \sum_{n=1}^{N_{all}} \sum_{i=1}^{E} p_{\mathrm{moe}}(t_n)_{id_{\mathrm{moe},n}},$$

(13)

where $N_{all}$ is the count of all *conflicting tokens*, $E$ is the number of experts, and $p_{\mathrm{moe}}(t_n)$ represents the routing score for the *conflicting token* $t_n$. $id_{\mathrm{moe},n}$ is the current expert ID of the *conflicting token* $t_n$. $(ii)$ CE-like: Utilizing the inverted routing score along with cross-entropy loss. Our motivation

Table 9: **Study about different conflict elimination loss designs.** The configure MoE-LLaVA-4Top2 with StableLM-1.6B is set for experiments.

| | VQA$^{v2}$ | GQA | VisWiz | SQA$^I$ | VQA$^T$ | POPE | MME | MMB | MM-Vet | Avg |
|---|---|---|---|---|---|---|---|---|---|---|
| MSE-like | 76.7* | 60.7* | 37.0 | 62.8 | 50.6 | 85.7 | 1346.5 | 60.6 | 27.8 | 57.7 |
| CE-like | 76.9* | 60.9* | 37.7 | 62.6 | 50.7 | 85.9 | 1355.1 | 60.7 | 28.2 | 58.0 |

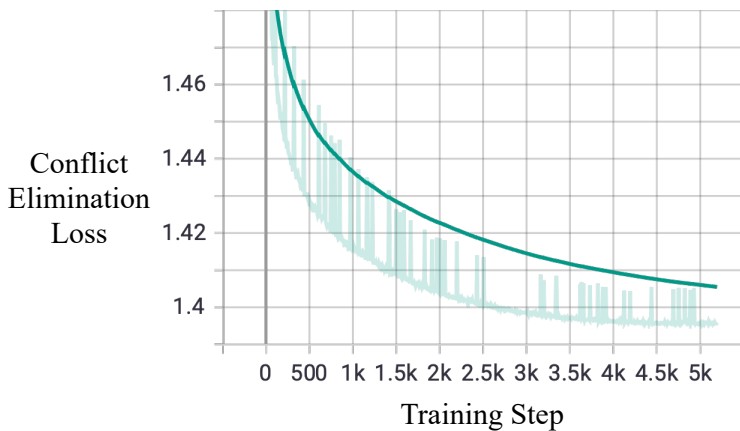

Figure 6: **Loss curve of conflict elimination loss**. The above graph shows the loss curve of conflict elimination loss during the normal training process when Phi2-2.7B is used as the LLM. Since the total number of sampling points is limited to 1000 in TensorBoard, the sampling interval is set to 7.

for taking the inverted routing score is to minimize the routing score of token on its current expert.

$$
\begin{aligned}
z'_{\text{moe}}(t_n) &= -z_{\text{moe}}(t_n), \\
p'_{\text{moe}}(t_n)_i &= \frac{e^{z'_{\text{moe}}(t_n)_i}}{\sum_{j=1}^{E} e^{z'_{\text{moe}}(t_n)_j}}, \\
\mathcal{L}_{\text{CEL}} &= \frac{1}{N_{all} \cdot E} \sum_{n=1}^{N_{all}} \sum_{i=1}^{E} \log(p'_{\text{moe}}(t_n)_i) \cdot q_{\text{moe}}(t_n)_i.
\end{aligned}
\tag{14}
$$

As shown in Table 9, the CE-like loss performs better. The reason may be that, although the optimization direction of the MSE-like loss is consistent with the CE-like loss, the optimization speed of the CE loss is superior to the MSE loss (Zhang & Sabuncu, 2018; Wang et al., 2019; Yang et al., 2020). An analysis of the gradient of CE (Yang et al., 2020) reveals that when the probability of the sample on the ground-truth class is small, CE will produce a significantly larger gradient than MSE.

**Loss Curve**: Different from the statistical verification in Figure 3 that only uses conflict elimination loss $\mathcal{L}_{\text{moe}}$, $\mathcal{L}_{\text{CEL}}$, $\mathcal{L}_{\text{moe}}$, and $\mathcal{L}_{\text{aux}}$ are used during the normal training process. We present the loss curve of the conflict elimination loss during training. As shown in Figure 6, the loss is convergent.

### 6.3.3 TOKEN GRADIENT STATISTIC

**Similarity statistic**: We compute the distribution of cosine similarity between the gradient of each token and the averaged gradient. Specifically, we randomly sample some data. We extract token-level gradients, calculate the average gradient $\mathbf{g}_{mean}$ for each expert, and compute the cosine similarity between the gradient of each token $\mathbf{g}_n$ and the average gradient of its current expert $\mathbf{g}_{mean}$. We perform the statistics for both initial and fully-trained models. As shown in Figure 7, we find that: $(i)$ In the initial model, there is a conflict between token gradients and average gradients; $(ii)$ In the fully-trained model, the conflict between token gradients and average gradients is reduced. This verifies that STGC can effectively increase the cosine similarity between the gradients of tokens within an expert and the average gradient.

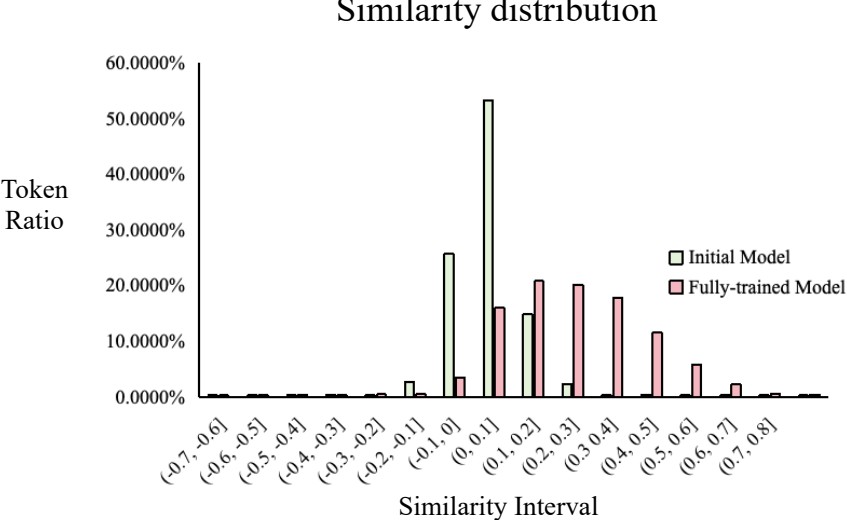

Figure 7: **Gradient similarity distribution**. We compute the distribution of cosine similarity between the gradient of each token and the averaged gradient. The configure MoE-LLaVA-4Top2 with StableLM-1.6B is set for experiments. We add the proposed STGC to train models, to generate a fully-trained model from the initial model.

***Gradient consistency std* and conflicting ratio**: We further explore the std deviation of *gradient consistency* and the ratio of *conflicting tokens*. Expanding on the statistical verification in the main part, we conduct a statistical verification experiment and define three metrics: The first step is to gather gradients of all tokens within an expert $e_i$ and compute their cosine similarity to form a similarity matrix. Then, we define the mean of the similarity matrix as $sim_i$. We define the mean of $sim_i$ on all experts as *gradient consistency*, serving as a novel metric to evaluate whether the gradient directions are consistent within the expert. We define the std deviation of $sim_i$ on all experts as *gradient consistency std*, serving as a metric to evaluate the degree of discreteness in the *gradient consistency* among different experts. Additionally, we calculate the number of *conflicting tokens* within all experts as $N_1$, and the total number of tokens as $N$, defining the conflicting ratio as $\frac{N_1}{N}$. The second step is to update models. We only use the proposed loss to update parameters, to undisturbedly observe its impact on three metrics.

We present the curve graph of the three metrics obtained from the TensorBoard dashboard in Figure 8. We can observe: $(i)$ as the conflict elimination loss decreases, the *gradient consistency* increases and the conflicting ratio decreases. This means that STGC effectively reduces gradient conflicts within the expert and reduces the count of *conflicting tokens*. $(ii)$ The *gradient consistency std* increases, meaning that the difference of *gradient consistency* among different experts enlarges. We speculate that this is due to the different rates at which *gradient consistency* increases across various layers. For example, Figure 3 (b) indicates that deeper layers have faster *gradient consistency* increase rate after adding STGC. Figure 9 also shows that STGC mainly reduces *conflicting tokens* at deep layers, and most *conflicting tokens* emerge in the shallow layers after learning.

**Conflicting token after learning**: In Figure 7, although we observe a significant reduction in *conflicting tokens* for the fully-trained model, we find that there are still some *conflicting tokens*. As shown in Figure 9, we analyze the layers in which they appear and find that most *conflicting tokens* emerge in the shallow layers after learning.

### 6.3.4 ROUTING

SMoE (Jiang et al., 2024) claims that "Surprisingly, we do not observe obvious patterns in the assignment of experts based on the topic." As shown in Figure 5, we also observe that the distribution of expert loading across some different datasets is similar, but we notice diversity in token-level activated pathways for different datasets. We suspect that the distribution of expert loading may not

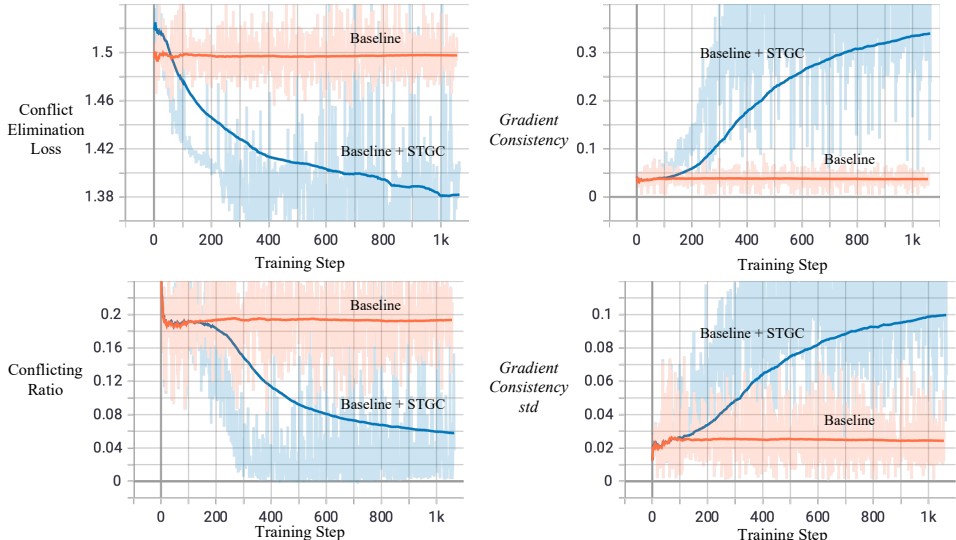

Figure 8: *Gradient consistency* **and conflicting ratio analysis**. The configure MoE-LLaVA-4Top2 with StableLM-1.6B is set for experiments. We finish the statistic on one GPU, so the total step number is 41581. The sampling interval is set to 1 in TensorBoard for the above graph and the sampling interval is 7 in TensorBoard for Figure 3, so the above graph appears to have a slight difference with Figure 3.

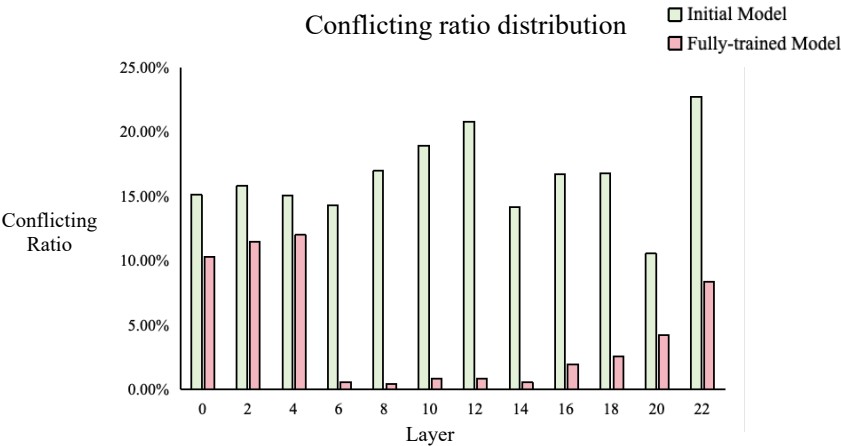

Figure 9: **Conflicting ratio distribution** on different MoE layers after learning.

be sufficiently accurate to reflect the routing of diverse data. Then, V-MoE (Riquelme et al., 2021) and MoNE (Jain et al., 2024) focus on leveraging the token importance difference to further accelerate MoE. V-MoE proposes Batch Prioritized Routing to discard unimportant tokens and MoNE proposes Expert Preferred Router to allocate more tokens to experts with a larger volume. We focus on avoiding token interference during training to enhance performance, which is parallel to the focus of V-MoE or MoNE. Thus, theoretically, STGC could be integrated with V-MoE or MoNE. Since MoNE does not have the official open-source code, we attempt to use Batch Prioritized Routing from V-MoE for further inference acceleration.

During inference, the eval capacity of MoE is set to 2.0 Lin et al. (2024). As shown in Table 10, when we reduce the capacity from 2.0 to 0.5, the model performance of both MoE-LLaVA and MoE-LLaVA+STGC declines significantly because many tokens are discarded. When Batch Prioritized Routing is added, there is a noticeable performance improvement. We find that when the capacity is reduced to 0.5, regardless of whether Batch Prioritized Routing is added or not, MoE-LLaVA+STGC

Table 10: **Study about Batch Prioritized Routing (BPR).** The configure MoE-LLaVA-4Top2 with StableLM-1.6B is set for experiments. BPR refers to Batch Prioritized Routing.

| | eval capacity | BPR | GQA | SQA$^\text{I}$ | VQA$^\text{T}$ | POPE | MME | MMB | MM-Vet | Avg |
|---|---|---|---|---|---|---|---|---|---|---|
| MoE-LLaVA | 2.0 | | 60.3$^*$ | 62.6 | 50.1 | 85.7 | 1318.2 | 60.2 | 26.9 | 57.6 |
| +STGC | 2.0 | | 60.9$^*$ | 62.6 | 50.7 | 85.9 | 1355.1 | 60.7 | 28.2 | 58.2 |
| MoE-LLaVA | 0.5 | | 10.2$^*$ | fail | 15.2 | 69.8 | fail | 6.0 | 16.4 | - |
| +STGC | 0.5 | | 21.1$^*$ | 18.0 | 13.6 | 71.9 | fail | 7.0 | 20.7 | - |
| MoE-LLaVA | 0.5 | ✓ | 51.0$^*$ | fail | 33.4 | 83.8 | 1032.4 | 26.8 | 22.0 | - |
| +STGC | 0.5 | ✓ | 58.0$^*$ | 57.7 | 44.3 | 85.3 | 1234.3 | 48.5 | 22.8 | 52.8 |

shows a significant performance improvement compared to MoE-LLaVA. A possible reason is that STGC can prevent a single expert from handling too many tokens, thereby reducing the number of discarded tokens.

## 6.4    COMPUTATIONAL OVERHEAD

STGC does not increase the inference overhead, while it may need the memory and time overhead during training. The memory overhead mainly results from storing gradients. The time overhead mainly results from computing gradients. We analyze the additional overhead during training from these two aspects.

**Training memory overhead**.   We begin our analysis from StableLM-1.6B. The FFN layer in StableLM-1.6B contains three linear layers, $fc^{gate} \in R^{2048 \times 5632}$, $fc^{up} \in R^{2048 \times 5632}$, and $fc^{down} \in R^{5632 \times 2048}$. As stated in section 6.1, for each token, we only store the gradient it produces on the bias within the experts. The token count per layer is about 1000, and each token only goes through one expert, so the gradient matrix stored per layer is $\mathbf{G} \in R^{1000 \times (5632+5632+2048)}$. We need to store a gradient matrix $\mathbf{G}$ for each MoE layer, and the MoE layer in StableLM-1.6B is $\{0, 2, 4, 6, 8, 10, 12, 14, 16, 18, 20, 22\}$ (the total layer number is 24). Thus, the theoretical memory overhead is $12 \times 1000 \times (5632+5632+2048)$. The parameter is bfloat16 (2 bytes), so the theoretical memory overhead is about 0.29 GB, which is significantly less than the memory cost of LLM and data (usually larger than 10 GB during training). The FFN layer in Phi2-2.7B contains two linear layers, $fc^1 \in R^{2560 \times 10240}$ and $fc^2 \in R^{10240 \times 2560}$, and Phi2-2.7B has 16 MoE layers. Thus, the theoretical memory overhead is $16 \times 1000 \times (2560 + 10240)$, *i.e.*, 0.38 GB.

If storing the token-level gradients on each weight, the required storage overhead is $12 \times 1000 \times (2048 \times 5632 + 2048 \times 5632 + 5632 \times 2048)$ (773 GB) for StableLM-1.6B and $16 \times 1000 \times (2560 \times 10240 + 10240 \times 2560)$ (1562 GB) for Phi2-2.7B, which is amazingly large. Thus, it is impossible to use the token-level gradient on each weight to conduct experiments.

**Training time overhead**. MoE-LLaVA performs a forward pass, followed by a backward pass to update parameters. As Section section 6.1 mentioned, MoE-LLaVA+STGC freezes parameters except for the bias within the experts, performs a forward pass, followed by a backward pass to compute token-level gradients; then, MoE-LLaVA+STGC unfreeze parameters and performs a backward pass to update parameters. Thus, the main time overhead results from "the computation of token-level gradients". We directly report the train_samples_per_second and train_steps_per_second recorded in "trainer_state.json" after training. As shown in Table 11, we have reduced the additional time overhead to about 20% through some engineering tricks (*e.g.*, only computing the token-level gradient on the bias and freezing parameters that do not require gradient computation). Some methods may further speed up STGC, such as using STGC only on even iterations or applying STGC to only half of the MoE layers. We will further explore these experiments in the future.

Besides, the overhead in gradient computation is a common issue faced by existing gradient-based methods. We believe that it is hopeful to address this common issue in the future, such as through gradient estimation methods (Mu et al., 2020; Paulus et al., 2020; Baydin et al., 2022; Shi et al., 2022; Wu et al., 2023).

Table 11: **Study about computational overhead.** We set MoE-LLaVA-4Top2 with StableLM-1.6B and Phi2-2.7B to conduct the analysis. One step (s) means the time needed for one step during training. STGC-full means computing token-level gradients on the weight.

| LLM | | Memory | | Time | | |
|---|---|---|---|---|---|---|
| | | theoretical overhead | train_samples _per_second | train_steps _per_second | one step (s) | |
| StableLM-1.6B | MoE-LLaVA | - | 19.796 | 0.154 | 6.494 | |
| | +STGC | 0.29 GB | 16.283 | 0.127 | 7.874 (+21.3%) | |
| | +STGC-full | 773 GB | fail | fail | fail | |
| Phi2-2.7B | MoE-LLaVA | - | 10.765 | 0.084 | 11.905 | |
| | +STGC | 0.38 GB | 8.851 | 0.069 | 14.493 (+21.7%) | |
| | +STGC-full | 1562 GB | fail | fail | fail | |

Table 12: **Study about language tasks.** Our study investigates the MoE integrated with STGC on language tasks using the GLUE benchmark (Wang, 2018), with BERT-large as the backbone model. MoE-8Top2 means a traditional MoE, configured with 8 experts, of which the Top-2 are activated, *i.e.*, 8Top2. * means the re-implemented results.

| | COLA | MRPC | QNLI | MNLI | RTE | Avg |
|---|---|---|---|---|---|---|
| MoE-8Top2 (Guo et al., 2024) | 64.5 | 90.2 | 92.4 | 86.7 | 74.9 | 81.7 |
| DYNMOE (Guo et al., 2024) | 65.2 | 90.6 | 92.6 | 86.4 | 73.4 | 81.6 |
| MoE-8Top2* | 64.5 | 90.0 | 93.4 | 86.9 | 72.9 | 81.5 |
| +STGC | 66.8 | 91.2 | 93.8 | 87.6 | 74.7 | 82.8 |

## 6.5 LANGUAGE TASKS

We study the use of STGC on language tasks. Theoretically, the deployment of STGC is not constrained to specific task types. To validate the generalization of STGC, we extend its use to language tasks. Specifically, we follow DYNMOE Guo et al. (2024) to apply the MoE framework for language tasks. Specifically, the language tasks adhere to the same settings as those in MoEfication Zhang et al. (2021) and EMoE (Qiu et al., 2023). The MoE is built upon the BERT-large Devlin (2018) architecture, employing the MoEfication method, and is fine-tuned on GLUE Wang (2018) benchmark, which encompasses COLA Warstadt (2019), QNLI Wang (2018), RTE Bentivogli et al. (2009), MNLI Xu et al. (2020), and MRPC Dolan & Brockett (2005). For MoE configuration, we set the total count of experts to 8, with the Top-2 experts being activated, and we refer to this configuration as 8Top2. We add the proposed STGC to the MoE. As shown in Table 14, the experimental results show the significant effectiveness of STGC on language tasks.

## 6.6 THEORETICAL ANALYSIS

We conduct a brief theoretical analysis based on the theory of PCGrad (Yu et al., 2020). Suppose there are tokens $t_n$ and $t'_n$, which pass through the same expert and generate gradients $\mathbf{g_n}$ and $\mathbf{g_{n'}}$ on that expert. Let $\mathbf{g_n} = \nabla \mathcal{L}_n$, $\mathbf{g_{n'}} = \nabla \mathcal{L}_{n'}$, and $\mathbf{g} = \nabla \mathcal{L} = \mathbf{g_n} + \mathbf{g_{n'}}$ ($\nabla \mathcal{L} = \nabla_\theta \mathcal{L}$, where $\theta$ is the parameter). $cos(\phi_{nn'})$ is the cosine similarity between gradients $\mathbf{g_n}$ and $\mathbf{g'_n}$. Token gradient conflicts mean the cosine similarity $cos(\phi_{nn'}) < 0$, $cos(\phi_{nn'}) < 0$ potentially leads to an increase in the loss. When $cos(\phi_{nn'}) > 0$, the loss decreases strictly, *i.e.*, $\nabla \mathcal{L} < 0$, which can reach the optimal value. Then, different from PCGrad, STGC does not alter the gradients of the tokens but changes the routing of the tokens to avoid conflicting gradients, with the function of increasing the cosine similarity $cos(\phi_{nn'})$ to satisfy the condition $cos(\phi_{nn'}) > 0$.

The notation $||\cdot||$ represents the $L_2$-norm. During each iteration, if $cos(\phi_{nn'}) > 0$, a standard gradient descent step with a learning rate $t \leq \frac{1}{L}$ is employed. This results in a strict reduction in the value of the objective function $\mathcal{L}(\theta)$, given that the function is convex, unless the gradient $\nabla \mathcal{L}(\theta)$ is zero, which happens exclusively when $\theta$ equals the optimal value $\theta^*$ (Boyd & Vandenberghe, 2004).

We further analyze the loss: Assuming that the gradient of the loss $\nabla\mathcal{L}$ is Lipschitz continuous with a Lipschitz constant $L$, it implies that the Hessian matrix $\nabla^2\mathcal{L}(\theta) - LI$ is negative semi-definite. Leveraging this property, we can perform a quadratic expansion of $\mathcal{L}$ around $\mathcal{L}(\theta)$, leading to the subsequent inequality:

$$\mathcal{L}(\theta^+) \leq \mathcal{L}(\theta) + \nabla\mathcal{L}(\theta)^T(\theta^+ - \theta) + \frac{1}{2}\nabla^2\mathcal{L}(\theta)||\theta^+ - \theta||^2$$

$$\leq \mathcal{L}(\theta) + \nabla\mathcal{L}(\theta)^T(\theta^+ - \theta) + \frac{1}{2}L||\theta^+ - \theta||^2$$

Then, given $\theta^+ = \theta - t \cdot \mathbf{g}$, the inequality can be expressed as:

$$\mathcal{L}(\theta^+) \leq \mathcal{L}(\theta) - t \cdot \mathbf{g}^T\mathbf{g} + \frac{1}{2}Lt^2||\mathbf{g}||^2$$

(Expanding, using the identity $\mathbf{g} = \mathbf{g_n} + \mathbf{g_{n'}}$)

$$= \mathcal{L}(\theta) - (t - \frac{1}{2}Lt^2)(||\mathbf{g_n}||^2 + ||\mathbf{g_{n'}}||^2 + 2 \cdot \mathbf{g_n} \cdot \mathbf{g_{n'}})$$

(Using the identity $\cos(\phi_{nn'}) = \dfrac{\mathbf{g_n} \cdot \mathbf{g_{n'}}}{||\mathbf{g_n}||||\mathbf{g_{n'}}||}$)

$$= \mathcal{L}(\theta) - (t - \frac{1}{2}Lt^2) \cdot (||\mathbf{g_n}||||\mathbf{g_{n'}}||)(\frac{||\mathbf{g_n}||}{||\mathbf{g_{n'}}||} + \frac{||\mathbf{g_{n'}}||}{||\mathbf{g_n}||} + 2 \cdot \cos(\phi_{nn'})).$$

$$(15)$$

By setting $t \leq \frac{1}{L}$, we can obtain $-(1 - \frac{1}{2}Lt) = \frac{1}{2}Lt - 1 \leq \frac{1}{2}L(1/L) - 1 = \frac{-1}{2}$ and $Lt^2 \leq t$.

Incorporating the bound into the previous expression, we can deduce the following conclusion:

$$\mathcal{L}(\theta^+) \leq \mathcal{L}(\theta) - \frac{1}{2}t(||\mathbf{g_n}||||\mathbf{g_{n'}}||)(\frac{||\mathbf{g_n}||}{||\mathbf{g_{n'}}||} + \frac{||\mathbf{g_{n'}}||}{||\mathbf{g_n}||} + 2 \cdot \cos(\phi_{nn'})).$$

If $\cos(\phi_{nn'}) < -\frac{1}{2} \cdot (\frac{||\mathbf{g_n}||}{||\mathbf{g_{n'}}||} + \frac{||\mathbf{g_{n'}}||}{||\mathbf{g_n}||})$, $\frac{||\mathbf{g_n}||}{||\mathbf{g_{n'}}||} + \frac{||\mathbf{g_{n'}}||}{||\mathbf{g_n}||} + 2 \cdot \cos(\phi_{nn'})$ will be negative. The bound of $\mathcal{L}(\theta^+)$ is larger than $\mathcal{L}(\theta)$, so the loss may increase. If $\cos(\phi_{nn'}) > 0$, $\frac{||\mathbf{g_n}||}{||\mathbf{g_{n'}}||} + \frac{||\mathbf{g_{n'}}||}{||\mathbf{g_n}||} + 2 \cdot \cos(\phi_{nn'})$ will always be positive. This positivity ensures that the objective function value decreases strictly with each iteration, suggesting that by repeatedly applying this process, we can converge to the optimal value. Note that this result only holds when we select a sufficiently small learning rate, specifically $t \leq \frac{1}{L}$.

The goal of STGC is to increase $cos(\phi_{nn'})$ to satisfy the condition $cos(\phi_{nn'}) > 0$. Figure 3 and Figure 8 have verified that STGC can increase $cos(\phi_{nn'})$. Consequently, STGC is beneficial to the convergence of the objective function towards its optimal value, thereby improving the overall effectiveness of the model.

## 6.7 GRADIENT CONFLICTING PHENOMENON

### 6.7.1 FEATURE-GRADIENT RELATIONSHIP

Similar features do not necessarily imply similar gradients (Mu et al., 2020). When tokens have highly similar features (similar tokens), the router can assign them to one expert. However, they may have dissimilar gradients. STGC can be understood as learning token features based on token-level gradient relationships. After using STGC, the features of tokens become less similar when they have dissimilar (conflicting) gradients, for being routed to different experts. To verify this, we sample some data for analysis. Suppose there are $N$ tokens and the cosine similarity is $S \in R^{N \times N}$ between features of these tokens. We flatten $S \in R^{N \times N}$ to $\bar{S} \in R^{N^2}$. Meanwhile, we calculate the cosine similarity $S_g \in R^{N \times N}$ between gradients of these tokens, flattening it to $\bar{S}_g \in R^{N^2}$. We compute

Table 13: **Study about feature-gradient relationship.** The configure MoE-LLaVA-4Top2 with StableLM-1.6B is set for experiments.

| | Pearson correlation coefficient |
|---|---|
| MoE-LLaVA | 0.2654 |
| + STGC | 0.3563 |

Table 14: **Study about gradient conflicting.** $z'_{i,y_i}$ denotes the average logit of tokens for their corresponding labels when calculating the main loss. $z'_{i,y_j}$ means the average logit of tokens on the labels of their conflicting tokens (meaning tokens having similar feature but divergent gradients). $z'_{i,o}$ denotes the average logit of tokens for all other labels, excluding the aforementioned labels.

| | $z'_{i,y_i}$ | $z'_{i,y_j}$ | $z'_{i,o}$ |
|---|---|---|---|
| Value | 23.4885 | 14.5673 | -0.6793 |

the Pearson correlation coefficient between $\bar{S}$ and $\bar{S}_g$ to measure whether feature relationships reveal gradient relationships. We select MoE-LLaVA and MoE-LLaVA + STGC for our experiments. As shown in Table 13, the experimental results indicate that token features and gradients have a higher correlation after using STGC.

### 6.7.2 POSSIBLE REASON FOR TOKEN GRADIENT CONFLICTING

In theory, similar tokens will be assigned to the same expert by a feature-based router. We focus on token gradient conflicting within an expert, so we want to further conduct a study about the phenomenon "similar tokens have divergent gradients". In specific, we compute the cosine similarities between token features in an expert as $S$. We then compute the cosine similarities between token gradients in an expert as $S_g$. We directly compute the difference $S' = S - S_g$. When $S'_{i,j}$ is large, token $t_i$ and token $t_j$ has a high feature similarity but a low gradient similarity, meaning that the phenomenon "similar tokens have divergent gradients" is significant. We find that $S'_{i,j}$ is large when two tokens are confusing. Suppose labels of two tokens $t_i$ on $y_i$ are $y_i$ and $y_j$ respectively. $z$ is the logit of each token for computing the main loss. $z_{i,y_i}$ means the logit of token $t_i$ on $y_i$. We define that $t_i$ and $t_j$ are confusing when $z_{i,y_j}$ and $z_{j,y_i}$ are high.

For example, in one expert, $S'$ has the maximum at $S'_{78,51}$. $S_{78,51}$ is 0.9316 and $S'_{78,51}$ is -0.6875. $t_{78}$ has the label 13 and $t_{51}$ has the label 11. We find that $t_{78}$ ($t_{51}$) has a second highest logit on 11 (13). This is somewhat similar to the class confusion phenomenon in traditional visual classification tasks. For example, in visual classification, dog and cat are visually similar and they have similar features, but the learning of cat may lead to the misclassification of dog.

We further sample some pairs of tokens to conduct a statistical verification. Each pair of tokens $\{t_i, t_j\}$ has the significant feature and gradient cosine similarity difference $S'_{i,j}$. $z_{i,y_i}$ means the logit of token $t_i$ on $y_i$. In addition, removing $z_{i,y_i}$ and $z_{i,y_j}$, we record the mean logit in $z_i$ as $z_{i,o}$. $z'_{i,y_j}$ is the average of $z_{i,y_i}$. As shown in the below table, $z'_{i,y_i}$ is significantly higher than $z'_{i,o}$, and close to $z'_{i,y_i}$, which validates the above hypothesis.

