# Supplementary Material about Solving Token Gradient Conflict in Mixture-of-Experts for Large Vision-Language Model

**Longrong Yang**[1], **Dong Shen**[2], **Chaoxiang Cai**[3], **Fan Yang**[2], **Tingting Gao**[2], **Di Zhang**[2], **Xi Li**[1,†]
[1]College of Computer Science and Technology, Zhejiang University
[2]Kuaishou Technology
[3]School of Software Technology, Zhejiang University

In our supplementary material, we provide the following details and experiments:

- Sec. A: We provide more engineering implementation details about training.
- Sec. B: We provide more implementation details about sample-level routing.
- Sec. C: We provide more experimental results about expert loading, loss design, token gradient statistic, and different routing mechanisms.
- Sec. D: We provide a analysis about computational overhead.
- Sec. E: We provide experimental results about language tasks.
- Sec. F: We provide a brief theoretical analysis.