# OpenReview forum: "Solving Token Gradient Conflict in Mixture-of-Experts for Large Vision-Language Model"
_ICLR.cc/2025/Conference — ICLR 2025 Poster_

### Official Review · Reviewer_BTYK · 2024-10-19

**Soundness:** 3
**Presentation:** 3
**Contribution:** 3
**Rating:** 6
**Confidence:** 4

**Summary:**

This paper proposes a novel approach to address token interference in Mixture-of-Experts (MoE) architectures for Large Vision-Language Models (LVLMs). While MoE improves computational efficiency by activating only a subset of experts during inference, token interference within the same expert remains a challenge. The authors introduce Solving Token Gradient Conflict (STGC), which leverages token-level gradient selection to detect conflicting tokens and implements a conflict elimination loss to optimize token routing, thereby mitigating interference. The proposed method serves as a plug-and-play solution for various LVLM architectures. Empirical results substantiate the effectiveness of the approach, demonstrating significant improvements across multiple datasets.

**Strengths:**

* **Novel Perspective on Token-Level Interference:** The paper introduces an insightful perspective on token interference by analyzing conflicting optimization directions among tokens assigned to the same expert, which is an important and under-explored issue.
* **Direct and Effective Methodology:** The method efficiently identifies conflicting tokens through comparisons between individual token gradients and the global average gradient within experts. The regularization loss improves routing by discouraging conflicting tokens from remaining with the same expert.
* **Comprehensive Experiments:** The experimental results are extensive and validate the proposed method’s effectiveness. The evaluations across diverse datasets highlight the robustness and general applicability of the approach.

**Weaknesses:**

* **Clarification on Gradient Conflict Hypothesis:** The authors assert that the learning of conflicting tokens increases the loss of most other tokens within the same expert. However, it is counterintuitive since conflicting tokens should be relatively few, making the directions of parameter optimization align with the global average gradient of all tokens in the expert. Clarifying this assumption would improve the paper's clarity.
* **Handling Conflicting Tokens Across Experts:** The paper suggests redistributing conflicting tokens to different experts. However, it is unclear how the method ensures that these conflicting tokens do not cause conflicts with tokens in other experts.
* **Questioning the Gating Network’s Learning Capabilities:** It remains unclear why the gating network fails to learn an optimal distribution for the conflicting tokens. Given that each expert should ideally handle tokens with similar characteristics, why would the gating network assign such divergent tokens to the same expert?

**Questions:**

All relevant questions are included under the “Weaknesses” section. If the authors can address these concerns adequately, I will consider increasing my rating.

---

> ### Author Response · Authors · 2024-11-23
> **Response to Reviewer BTYK**
>
> Thank you for your review and your feedback on our paper. We address your questions and concerns below.
>
> > _Comment 1: Clarification on Gradient Conflict Hypothesis. The authors assert that the learning of conflicting tokens increases the loss of most other tokens within the same expert. However, it is counterintuitive since conflicting tokens should be relatively few, making the directions of parameter optimization align with the global average gradient of all tokens in the expert. Clarifying this assumption would improve the paper's clarity._
>
> Response: Thank you for pointing out this.
> "The learning of _conflicting tokens_ increases the loss of most other tokens within the same expert" is not precise enough.
> We have modified it as "The learning of _conflicting tokens_ increases the average loss of tokens within their current experts" in the manuscript.
> We further explain this claim: the gradient generated by _conflicting token_ is in the opposite direction of the average gradient. Updating parameters along the direction of the _conflicting token_ gradient is equivalent to updating parameters in the direction of the average gradient's ascent, which leads to an increase in the average loss.
> For example, when an expert has _conflicting tokens_, the average loss decreases from 0.3 to 0.25 after updating the parameters; after removing _conflicting tokens_, the average loss may decrease from 0.3 to 0.2 after updating the parameters.
>
> > _Comment 2: Handling Conflicting Tokens Across Experts: The paper suggests redistributing conflicting tokens to different experts. However, it is unclear how the method ensures that these conflicting tokens do not cause conflicts with tokens in other experts._
>
> Response: It is challenging to determine the optimal expert of a _conflicting token_ during redistributing.
> We have recognized this limitation and presented it in the Limitation part.
> However, importantly, even if some tokens remain conflicting after redistributing, it is unlikely to be the case for all; STGC cannot guarantee the total elimination of _conflicting tokens_, but it can reduce the number of _conflicting tokens_.
> In Figure 3, Figure 7, and Figure 8 of the manuscript, it has been confirmed that STGC is beneficial for improving _gradient consistency_ and reducing _conflicting tokens_.
> In the future, exploring whether it is possible to determine the optimal expert of a token from the optimization perspective may be an intriguing direction.
>
> > _Comment 3: Questioning the Gating Network’s Learning Capabilities: It remains unclear why the gating network fails to learn an optimal distribution for the conflicting tokens. Given that each expert should ideally handle tokens with similar characteristics, why would the gating network assign such divergent tokens to the same expert?_
>
> Response: Similar features do not necessarily imply similar gradients.
> When tokens have highly similar features (similar tokens), the router will assign them to one expert.
> However, they may have dissimilar gradients.
> STGC can be understood as learning token features based on token-level gradient relationships.
> After using STGC, the features of tokens become less similar when they have dissimilar (conflicting) gradients, for being routed to different experts.
> To verify this, we compute the Pearson correlation coefficient between feature similarities and gradient similarities to measure whether feature relationships reveal gradient relationships.
> As shown in the below table, the experimental results indicate that token features and gradients have a higher correlation after using STGC.
> For more details please refer to **Section 6.8** and **Table 12** of the revised manuscript.
>
> |                 &nbsp;                 | Pearson correlation coefficient |
> |----------------------------------------|---------------------------------|
> | MoE-LLaVA                              | 0.2654                          |
> | + STGC                                  | 0.3563                          |

---

> > ### Comment · Reviewer_BTYK · 2024-11-25
> >
> > Thank you for your rebuttal. While some concerns have been addressed, they do not fully convince me regarding comments #2 and #3. Therefore, I will maintain my original rating of 6.

---

> > > ### Author Response · Authors · 2024-11-27
> > > **Response to Reviewer BTYK**
> > >
> > > Thank you for your further response! We sincerely appreciate the time and effort you've dedicated to our paper.
> > >
> > > We strive to address all your concerns thoroughly; therefore, we would like to provide further clarification on **_comment #3_**:
> > >
> > > _Why would the gating network assign such divergent tokens to the same expert?_
> > >
> > > Because the feature-based gating network assigns tokens based on their features, not concerning their gradients on experts. We agree that "each expert should ideally handle tokens with similar characteristics"， but these tokens may have divergent gradients on the same expert. The reason is that, **the feature is only determined by the current token, but the gradient is determined by the loss of the current token, and the loss of the current token depends on the next token** (label).
> > >
> > > **To gain a deeper understanding of the phenomenon "similar tokens have divergent gradients", we conduct an in-depth analysis and find that this phenomenon is especially significant when two tokens are confusing.**
> > > Suppose there are tokens $t_i$ and $t_j$,  and their corresponding labels (next tokens) are $y_i$ and $y_j$, respectively.
> > > $z$ is the output logit of each token used to compute the main loss.
> > > $z_{i,y_i}$ means the output logit of token $t_i$ on $y_i$.
> > > **We define $t_i$ and $t_j$ as confusing when $z_{i,y_j}$ and $z_{j,y_i}$ are high.**
> > > For example, we find that in one sample, the feature cosine similarity between $t_{78}$ and $t_{51}$ ($t_{78}$ and $t_{51}$ are on the same expert) is 0.9316, but their gradient cosine similarity is only -0.6875, where "similar tokens have divergent gradients" is significant.
> > > $t_{78}$ has the label 13 and $t_{51}$ has the label 11.
> > > We find that, apart from the logits for their label classes, $t_{78}$ ($t_{51}$) has the highest logit on 11 (13).
> > > **This is somewhat similar to the class confusion phenomenon in traditional visual classification tasks.**
> > > For example, in visual classification, _dog_ and _cat_ are visually similar and they have similar features, but the learning of _cat_ may lead to the misclassification of _dog_.
> > >
> > > We further sample some pairs of tokens to conduct a statistical verification.
> > > Each pair of tokens $\{t_i,t_j\}$ has a significant feature and gradient cosine similarity difference.
> > > $z_{i,y_i}$ means the logit of token $t_i$ on $y_i$.
> > > Meanwhile, removing $z_{i,y_i}$ and $z_{i,y_j}$, we record the mean logit in $z_{i}$ as $z_{i, o}$.
> > > $z_{i,y_j}'$ is the average of $z_{i,y_i}$.
> > > **As shown in the below table, $z_{i,y_i}'$ is significantly higher than $z_{i,o}'$ and close to $z_{i,y_i}'$, which validates the above hypothesis.**
> > >
> > > |           | $z_{i,y_i}'$ | $z_{i,y_j}'$ | $z_{i,o}'$ |
> > > |-----------|-------------------|-------------------|-----------------|
> > > | Value     | 23.4885           | 14.5673           | -0.6793          |
> > >
> > > For more details please refer to **Section 6.11** and **Table 14**.
> > >
> > > Regarding **_comment #2_**, since in each iteration, a token only passes one expert and not the others in each MoE layer, its gradient reflects whether it conflicts with the holistic optimization direction of its current expert but it is difficult to determine its relationship with other experts. This makes determining the optimal expert for a _conflicting token_ challenging.
> > > To address this, a mechanism needs to be established to determine the relationships between _conflicting tokens_ and other experts.
> > > For example, based on the above new finding, it is possible to estimate the confusion between the label of a _conflicting token_ and the labels of tokens in other experts by introducing the prior, _e.g._, the prior "_cat_ and _dog_ are confusing".
> > > Then, the _conflicting token_ should be redistributed to the expert with minimal confusion, for preventing conflict.
> > > We will release all the code to benefit further research in the community.

---

### Official Review · Reviewer_5uJQ · 2024-10-30

**Soundness:** 3
**Presentation:** 2
**Contribution:** 3
**Rating:** 6
**Confidence:** 4

**Summary:**

The paper introduces Solving Token Gradient Conflict (STGC), a method to address interference between tokens within experts in Mixture-of-Experts (MoE) models for Large Vision-Language Models (LVLMs). STGC uses token-level gradient analysis to identify conflicting tokens and a regularization loss to optimize token routing, reducing interference and improving model performance. Experiments show STGC's effectiveness across diverse datasets, especially with larger data diversity. The method serves as a plug-in for existing MoE-based LVLMs. The study highlights the importance of managing token interference in MoE architectures and provides a novel approach to enhance their performance in vision-language tasks.

**Strengths:**

1. Routing is the key problem when training MoE and this paper targets this important problem to reduce interference between tokens within an expert in MoE models by using token-level gradient analysis.
2. The method demonstrated significant performance improvements on various vision-language benchmarks, indicating its effectiveness in enhancing model capabilities.
3. STGC is designed as a plug-in, which means it can be easily integrated into existing MoE-based LVLMs without the need for fundamental architecture changes.

**Weaknesses:**

1. The paper primarily focuses on the empirical validation of the STGC method. While the experimental results are promising, there could be a more in-depth theoretical analysis to understand the underlying principles and limitations of token gradient conflicts and how STGC addresses them.
2. The paper mentions the reduction in inference cost but does not discuss the potential increase in training cost due to the additional computations required for token-level gradient analysis.
3. The paper demonstrates the effectiveness of STGC on vision-language tasks. However, it is not clear how well these findings generalize to other domains or tasks outside of vision-language models. Further experiments in diverse domains could strengthen the paper's claims.

**Questions:**

Can you provide an analysis of the training costs?

---

> ### Author Response · Authors · 2024-11-23
> **Response to Reviewer 5uJQ (Part 1)**
>
> Thank you for your review and your feedback on our paper. We address your questions and concerns below.
>
> > _Comment 1: The paper primarily focuses on the empirical validation of the STGC method. While the experimental results are promising, there could be a more in-depth theoretical analysis to understand the underlying principles and limitations of token gradient conflicts and how STGC addresses them._
>
> Response: Thanks for your advice.
> We have conducted a brief theoretical analysis in **Section 6.9** of the revised manuscript.
>
> Suppose there are tokens $t_n$ and $t_n'$, which pass through the same expert and generate gradients $\mathbf{g_n}$ and $\mathbf{g_n'}$ on that expert.
> $cos(\phi_{nn'})$ is the cosine similarity between gradients $\mathbf{g_n}$ and $\mathbf{g_n'}$.
> Token gradient conflicts mean the cosine similarity $cos(\phi_{nn'}) < 0$,
> We can prove that $cos(\phi_{nn'}) < 0$ potentially leads to an increase in the loss.
> We can prove that when $cos(\phi_{nn'})$>0, the loss decreases strictly, which can reach the optimal value.
> The goal of STGC is to increase $cos(\phi_{nn'})$ to satisfy the condition $cos(\phi_{nn'})$>0.
> Figure 3 and Figure 8 verify that STGC increases $cos(\phi_{nn'})$.
>
> > _Comment 2: The paper mentions the reduction in inference cost but does not discuss the potential increase in training cost due to the additional computations required for token-level gradient analysis._
>
> Response: We have conducted a thorough computational overhead analysis in **Section 6.6** and **Table 10** of the revised manuscript.
>
> **Training memory overhead**.
> We compute the memory overhead by the size of the stored gradient.
> As stated in Section 6.1, for each token, we only store the gradient it produces on the bias within the experts. Thus, the theoretical memory overhead is merely about 0.29 GB on StableLM-1.6B and 0.38 GB on Phi2-2.7B, which is negligible during the training of MoE-based LVLMs. For the specific calculation please refer to Section 6.6.
>
> **Training time overhead**.
> We directly report the train_samples_per_second and train_steps_per_second recorded in "trainer_state.json" after training.
> MoE-LLaVA performs a forward pass, followed by a backward pass to update parameters.
> As Section 6.1 mentioned, MoE-LLaVA+STGC freezes parameters except for the bias within the experts, performs a forward pass, followed by a backward pass to compute token-level gradients; then, MoE-LLaVA+STGC unfreeze parameters and performs a backward pass to update parameters.
> Thus, the main time overhead results from "the computation of token-level gradients".
> As shown in the below tables, we have reduced the additional time overhead to about 20\% through some engineering tricks (_e.g._, only computing the token-level gradient on the bias and freezing parameters that do not require gradient computation).
> Some other methods may further speed up STGC, such as using STGC only on even iterations or applying STGC to only half of the MoE layers.
> We will further explore these experiments in the future.
>
> Computational overhead analysis on StableLM-1.6B:
> | Method      |  (theoretical memory overhead) | (train_samples_per_second) | (train_steps_per_second) | (one step (s))       |
> |-------------|-----------------------------|--------------------------|------------------------|--------------------|
> | MoE-LLaVA   | -                           | 19.796                   | 0.154                  | 6.494              |
> | +STGC       | 0.29 GB                     | 16.283                   | 0.127                  | 7.874 (+21.3%)     |
> | +STGC-full  | 773 GB                    | fail                     | fail                   | fail               |
>
> (_STGC-full means computing token-level gradients on the weight.
> Meanwhile, STGC is the used scheme, only computing token-level gradients on the bias._)
>
> Computational overhead analysis on Phi2-2.7B:
> | Method      | (theoretical memory overhead) | (train_samples_per_second) | (train_steps_per_second) | (one step (s))       |
> |-------------|-------------------------------|-----------------------------|---------------------------|----------------------|
> | MoE-LLaVA   | -                             | 10.765                      | 0.084                     | 11.905               |
> | +STGC       | 0.38 GB                       | 8.851                       | 0.069                     | 14.493 (+21.7%)      |
> | +STGC-full  | 1562 GB                       | fail                        | fail                      | fail                 |

---

> > ### Author Response · Authors · 2024-11-23
> > **Response to Reviewer 5uJQ (Part 2)**
> >
> > > _Comment 3: The paper demonstrates the effectiveness of STGC on vision-language tasks. However, it is not clear how well these findings generalize to other domains or tasks outside of vision-language models. Further experiments in diverse domains could strengthen the paper's claims._
> >
> > Response: Thanks for your advice.
> > Theoretically, the deployment of STGC is not restricted to task types.
> > To verify this, we extend STGC to language tasks.
> > We are still preparing the experimental results of this part, and we will report them within the next day or two.
> >
> > > _Comment 4: Can you provide an analysis of the training costs?_
> >
> > Response: Please refer to the response to _Comment 2_.

---

> ### Author Response · Authors · 2024-11-25
> **Response to Reviewer 5uJQ**
>
> We would like to update the experimental result below. Sorry for the late.
> > _Comment 3: The paper demonstrates the effectiveness of STGC on vision-language tasks. However, it is not clear how well these findings generalize to other domains or tasks outside of vision-language models. Further experiments in diverse domains could strengthen the paper's claims._
>
> Response: Thanks for your advice.
> Theoretically, the deployment of STGC is not constrained to specific task types.
> Thus, to validate the generalization of STGC, we extend its use to language tasks.
> We have supplemented the study on language tasks on **Section 6.10** and **Table 13**.
> We follow DYNMOE [1] to apply the MoE framework for language tasks and we add the proposed STGC to the MoE.
> As shown in the below table, the experimental results show the effectiveness of STGC on language tasks.
> For more details please refer to Section 6.10 and Table 13.
>
> |                |  COLA  |  MRPC  |  QNLI  |  MNLI  |  RTE   |  Avg  |
> |-----------------|--------|--------|--------|--------|--------|-------|
> | MoE-8Top2 [1] |  64.5  |  90.2  |  92.4  |  86.7  |  74.9  | 81.7 |
> | DYNMOE [1]   |  65.2  |  90.6  |  92.6  |  86.4  |  73.4  | 81.6 |
> | MoE-8Top2$^*$   |  64.5  |  90.0  |  93.4  |  86.9  |  72.9  | 81.5 |
> | +STGC           |  66.8  |  91.2  |  93.8  |  87.6  |  74.7  | 82.8 |
>
> $*$  means the re-implemented results.
>
> [1] Guo, Yongxin, et al. "Dynamic mixture of experts: An auto-tuning approach for efficient transformer models." arXiv preprint arXiv:2405.14297 (2024).

---

### Official Review · Reviewer_LbXo · 2024-11-02

**Soundness:** 3
**Presentation:** 3
**Contribution:** 3
**Rating:** 8
**Confidence:** 4

**Summary:**

The authors make a very interesting and useful observation regarding the inconsistency/randomness in gradient direction for tokens routed to a particular expert in the Mixture of Experts framework. They then propose adding a loss term that aims to align the expert-wise token gradients to the expert-wise mean gradient over all tokens. The results presented by the authors suggest that the gradient alignment correlates with better performance, and also provide ablations showing that their method actually gets the token gradients to align.

**Strengths:**

**Relevance**

Routing has been a difficult task, which is pertinent to the Mixture of Experts framework. However, the understanding and analysis of the router predictions has been lacking in the literature. I believe this paper takes a significant step towards identifying and addressing a potential issue that leads to hindrance in learning in the MOE framework. The authors define the notion of a conflicting token, and then addresses it well using their proposed method, comprehensively justifying the performance in a variety of scenarios.

**Presentation**

I enjoyed reading the paper, it was easy to follow (except some parts listed in weaknesses), introduces the problem well, and presents detailed results and ablations to justify their proposed method. I specifically appreciate the statistical verification in Figure 3 along with the results which indicates that token conflict is actually a problem and resolving it leads to better performance.

**Practicality**

The section on memory considerations and the engineering tricks for reducing memory overheads is very helpful and leads to a much practical implementation of the proposed method.

**Weaknesses:**

I will combine and list the weaknesses and questions here.

**Practical Implementation**: While sec A in the supplementary compares the practical implementation with the heavy one in terms of Pearson correlation coefficient between the similarity metrics, I believe additional considerations
- a quantitative measure of the memory overhead reduction (if not possible then a qualitative measure)
- performance difference on an actual dataset
could make the claim of practicality even stronger. Specific experiments or analysis in the main paper that quantifies the memory savings and any potential trade-offs in performance when using their practical implementation versus the full gradient computation would be helpful.

**Token Conflicts after STGC**: Fig 3 indicates the increase in gradient alignment over the course of the training process, but it would be interesting to understand the tokens that remain conflicting even after STGC. What proportion of the total tokens stay conflicting? Can something be reasoned about this behaviour? Along with the mean gradient consistency, it would be good to report the std deviations as well. It might also be worth tracking the proportion of total tokens that stay conflicting over the course of training. The authors' insights and justifications on this consideration would be useful.

**Conflict Elimination Loss**: Eqn 8 proposes the loss term, and the results do indicate its efficacy, but a more detailed motivation for the particular form of the loss could be useful. Specifically, why is $z_{moe}'(t_n)$ set to $-z_{moe}(t_n)$. Are there other forms that were considered, or insights into why this particular expression works would help.

**MOE Motivation**: The authors work with the assumption that MOE frameworks reduce interference between tokens from diverse data. However, there have been previous works that mention that it is not always the case. Particularly, [1] and [2] suggest that MoE does not necessarily lead to diverse data going to different experts, or observe weak correlation of router decisions with diversity. Additionally, [3] presents a MoE variant that works against this assumption, and uses nested experts. It would be helpful to suggest how STGC could be utilized in such scenarios. It might be useful to include a discussion section in the paper that explicitly addresses these alternative perspectives on MoE behavior
- how STGC relates to or differs from these other findings, and
- how STGC could be adapted or extended to work with different MoE variants like nested experts.

[1] Mixtral of Experts, Jiang et. al,, https://arxiv.org/abs/2401.04088

[2] Scaling Vision with Sparse Mixture of Experts, Riquelme et. al, https://arxiv.org/abs/2106.05974

[3] Mixture of Nested Experts: Adaptive Processing of Visual Tokens, Jain et. al, https://arxiv.org/abs/2407.19985

**Questions:**

Please see the weaknesses section for questions as well.

---

> ### Author Response · Authors · 2024-11-23
> **Response to Reviewer LbXo (Part 1)**
>
> Thank you for your review and your feedback on our paper. We address your questions and concerns below.
>
> > _Comment 1: Practical Implementation: A quantitative measure of the memory overhead reduction. Performance difference on an actual dataset could make the claim of practicality even stronger._
>
> Response: Thanks for your advice.
> We have conducted a detailed memory overhead analysis in **Section 6.6** and **Table 10** of the revised manuscript.
> We compute the memory overhead by the size of the stored gradient.
> As stated in Section 6.1, for each token, we only store the gradient it produces on the bias within the experts. Thus, the theoretical memory overhead is merely about 0.29 GB on StableLM-1.6B and 0.38 GB on Phi2-2.7B, which is negligible during the training of MoE-based LVLMs.
>
> If storing the token-level gradients on each weight, the required storage overhead is 773 GB for StableLM-1.6B and 1562 GB for Phi2-2.7B, which is amazingly large.
> Thus, it is impossible to use the token-level gradient on each weight to conduct experiments.
>
> For the specific calculation please refer to Section 6.6.
>
> Computational overhead analysis on StableLM-1.6B:
> | Method      |  (theoretical memory overhead) | (train_samples_per_second) | (train_steps_per_second) | (one step (s))       |
> |-------------|-----------------------------|--------------------------|------------------------|--------------------|
> | MoE-LLaVA   | -                           | 19.796                   | 0.154                  | 6.494              |
> | +STGC       | 0.29 GB                     | 16.283                   | 0.127                  | 7.874 (+21.3%)     |
> | +STGC-full  | 773 GB                    | fail                     | fail                   | fail               |
>
> (_STGC-full means computing token-level gradients on the weight.
> Meanwhile, STGC is the used scheme, only computing token-level gradients on the bias._)
>
> Computational overhead analysis on Phi2-2.7B:
> | Method      | (theoretical memory overhead) | (train_samples_per_second) | (train_steps_per_second) | (one step (s))       |
> |-------------|-------------------------------|-----------------------------|---------------------------|----------------------|
> | MoE-LLaVA   | -                             | 10.765                      | 0.084                     | 11.905               |
> | +STGC       | 0.38 GB                       | 8.851                       | 0.069                     | 14.493 (+21.7%)      |
> | +STGC-full  | 1562 GB                       | fail                        | fail                      | fail                 |
>
> > _Comment 2: Token Conflicts after STGC: Along with the mean gradient consistency, it would be good to report the std deviations as well. It might also be worth tracking the proportion of total tokens that stay conflicting over the course of training._
>
> Response: Thanks for your advice.
> We have reported the std deviation of _gradient consistency_ and the ratio of _conflicting tokens_ in **Section 6.5** and **Figure 8** of the revised manuscript.
> We can observe: $(i)$ as the conflict elimination loss decreases, the _gradient consistency_ increases, and the ratio of _conflicting tokens_ decreases.
> This means that STGC effectively reduces gradient conflicts within the expert and reduces the count of conflicting tokens.
> $(ii)$ The _gradient consistency std_ increases, meaning that the difference of _gradient consistency_ of different experts enlarges.
> We speculate that this is due to the different rates at which _gradient consistency_ increases at various layers.
> For example, Figure 3 (b) indicates that deeper layers have a faster _gradient consistency_ increase rate after adding STGC.
> Furthermore, we analyze the layers in which _conflicting tokens_ appear in **Figure 9** of the revised manuscript.
> We find that most _conflicting tokens_ emerge in the shallow layers after learning, further verifying the finding in Figure 3 (b) and Figure 8.

---

> ### Author Response · Authors · 2024-11-23
> **Response to Reviewer LbXo (Part 2)**
>
> > _Comment 3: Conflict Elimination Loss: Are there other forms that were considered, or insights into why this particular expression works would help._
>
> Response: Thanks for your advice.
> We have supplemented the discussion about conflict elimination loss design in **Section 6.4** and **Table 9** of the revised manuscript.
> In specific, the goal of the conflict elimination loss is to reduce the routing score $p_{\text{moe}}(t_n)$ of the _conflicting token_ $t_n$ on its current expert.
> We discuss different designs for the conflict elimination loss:
> $(i)$ MSE-like: Simply setting the routing score $p_{\text{moe}}(t_n)$ to the minimum.
> $(ii)$ CE-like: Utilizing the inverted routing score along with cross-entropy loss. Our motivation for taking the inverted routing score is to minimize the routing score of the token on its current expert.
> As shown in the below table, the CE-like loss performs better.
> The reason may be that, although the optimization direction of the MSE-like loss is consistent with the CE-like loss, the optimization speed of the CE loss is superior to the MSE loss [1,2].
>
> [1] Zhang, Zhilu, and Mert Sabuncu. "Generalized cross entropy loss for training deep neural networks with noisy labels." Advances in neural information processing systems 31 (2018).
>
> [2] Wang, Yisen, et al. "Symmetric cross entropy for robust learning with noisy labels." Proceedings of the IEEE/CVF international conference on computer vision. 2019.
>
> |                 &nbsp;                 | VQA$^\text{v2}$ | GQA | VisWiz | SQA$^\text{I}$ | VQA$^\text{T}$ | POPE | MME  | MMB | MM-Vet | Avg  |
> |------------------------------------------|-----------------|-----|--------|----------------|----------------|-------|------|------|--------|------|
> | MSE-like                                | 76.7$^*$        | 60.7$^*$ | 37.0  | 62.8            | 50.6            | 85.7  | 1346.5 | 60.6  | 27.8  | 57.7 |
> | CE-like                                  | 76.9$^*$        | 60.9$^*$ | 37.7  | 62.6            | 50.7            | 85.9  | 1355.1 | 60.7  | 28.2  | 58.0 |

---

> > ### Author Response · Authors · 2024-11-23
> > **Response to Reviewer LbXo (Part 3)**
> >
> > > _Comment 4: MOE Motivation: It might be useful to include a discussion section in the paper that explicitly addresses these alternative perspectives on MoE behavior._
> >
> > Response: Thanks for your advice.
> > We have provided the related discussion in Section **6.7** and **Table 11** of the revised manuscript.
> > Specifically, SMoE [1] claims that "Surprisingly, we do not observe obvious patterns in the assignment of experts based on the topic."
> > As shown in Figure 5 of the manuscript, we also observe that the distribution of expert loading across some different datasets is similar, but we notice diversity in token-level activated pathways for different datasets.
> > We suspect that the distribution of expert loading may not be sufficiently accurate to reflect the diverse token-level routing.
> > Then, V-MoE [2] and MoNE [3] focus on leveraging the token importance difference to further accelerate MoE.
> > V-MoE proposes Batch Prioritized Routing to discard unimportant tokens and MoNE proposes Expert Preferred Router to allocate more tokens to experts with a larger volume.
> > We focus on avoiding token interference during training to enhance performance, which is parallel to the focus of V-MoE or MoNE.
> > Thus, theoretically, STGC could be integrated with V-MoE or MoNE.
> > Since MoNE does not have the official open-source code, we attempt to use Batch Prioritized Routing from V-MoE for further inference acceleration.
> >
> > During inference, the eval capacity of MoE is set to 2.0.
> > As shown in the below table, when we reduce the capacity from 2.0 to 0.5, the model performance of both MoE-LLaVA and MoE-LLaVA+STGC declines significantly because many tokens are discarded.
> > When Batch Prioritized Routing is added, there is a noticeable performance improvement for both.
> > We find that when the capacity is reduced to 0.5, regardless of whether Batch Prioritized Routing is added or not, MoE-LLaVA+STGC shows a significant performance improvement compared to MoE-LLaVA.
> > A possible reason is that STGC can prevent a single expert from handling too many tokens, thereby reducing the number of discarded tokens.
> >
> > |               | eval capacity | BPR | GQA      | SQA I | VQA T | POPE | MME    | MMB  | MM-Vet | Avg  |
> > |---------------|---------------|-----|----------|-------|-------|------|--------|------|--------|------|
> > | MoE-LLaVA     | 2.0           |     | 60.3*    | 62.6  | 50.1  | 85.7 | 1318.2 | 60.2 | 26.9   | 57.6 |
> > | +STGC         | 2.0           |     | 60.9*    | 62.6  | 50.7  | 85.9 | 1355.1 | 60.7 | 28.2   | 58.2 |
> > | MoE-LLaVA     | 0.5           |     | 10.2*    | fail  | 15.2  | 69.8 | fail   | 6.0  | 16.4   | -    |
> > | +STGC         | 0.5           |     | 21.1*    | 18.0  | 13.6  | 71.9 | fail   | 7.0  | 20.7   | -    |
> > | MoE-LLaVA     | 0.5           | ✓   | 51.0*    | fail  | 33.4  | 83.8 | 1032.4 | 26.8 | 22.0   | -    |
> > | +STGC         | 0.5           | ✓   | 58.0*    | 57.7  | 44.3  | 85.3 | 1234.3 | 48.5 | 22.8   | 52.8 |

---

> ### Comment · Reviewer_LbXo · 2024-11-24
>
> Thank you for all your efforts in answering my concerns, the current version of the manuscript looks much more self-contained. I am happy to raise my score to 8.

---

> > ### Author Response · Authors · 2024-11-25
> > **Response to Reviewer LbXo**
> >
> > Thank you very much for your thoughtful feedback and for raising your score.
> > Your suggestions are invaluable in helping us improve the paper.
> > We sincerely appreciate your recognition of our efforts to address the major concerns.

---

> ### Public Comment · ~Sen_Pei1 · 2025-09-23
> **About the equation (8) in this paper.**
>
> It seems that the eq (8) is missing a negative sign. If I want to minimize the loss CEL, what I need to do is to minimize the logit of $z_{moe}(t_n)$ for conflicting token $t_n$ instead of minimizing $-z_{moe}(t_n)$.

---

> ### Public Comment · ~Longrong_Yang2 · 2025-09-23
>
> The standard CE loss increases the score of a sample on the ground-truth class.  In our case, we want to reduce the score of the sample on $id_{\text{moe},n}$.  Therefore, we add a negative sign, making our loss increase with $z_{\text{moe}}^\prime(t_n)$, which in turn decreases $z_{\text{moe}}(t_n)$.

---

### Official Review · Reviewer_EECo · 2024-11-03

**Soundness:** 2
**Presentation:** 2
**Contribution:** 2
**Rating:** 6
**Confidence:** 3

**Summary:**

This paper identifies conflicting token as one of the major problem in MoE-based LVLM training. Furthermore, the author propose a novel Conflicting-Elimination-Loss to resolve the conflict. By applying the proposed loss, it provides performance improvements over baseline.

**Strengths:**

(1) The proposed approach is novel and is effective in improving the model performance.
(2) Resolving conflicting-token demonstrates a strong correlation with improvement of the performance, which makes it a seemingly good metric to study MoE.
(3) Thorough empirical results are provided to illustrate the effectiveness of the method. Sensitivity analysis are provided to show the robustness of the method.

**Weaknesses:**

(1) Visualization on the routing mechanism of the trained model should be studied to provide insights into the trained model. By conducting such visualization, we might be able to know better why the proposed method can yield performance improvements. In addition, this could also validate the proposed method would still effectively utilize every one of the expert, instead of collapsing into only using one expert.
(2) It seems that the optimization process of the proposed CEL is not very stable (as in Figure 3(a)), the reviewer wonders whether this would be a problem when scaling up to larger scale.
(3) Computational overhead should be provided to ensure the proposed method does not incur too much overhead. This added study can show whether the proposed method is promising in terms of scaling up to large scale.

**Questions:**

See weakness section.

---

> ### Author Response · Authors · 2024-11-23
> **Response to Reviewer EECo (Part 1)**
>
> Thank you for your review and your feedback on our paper. We address your questions and concerns below.
>
> > _Comment 1: Visualization on the routing mechanism of the trained model should be studied to provide insights into the trained model._
>
> Response: Thanks for your advice.
> We have presented a visualization of the routing mechanism of the trained model in **Section 6.3** and **Figure 5** of the revised manuscript.
> Specifically, we follow MoE-LLaVA to obtain the distribution of expert loading and the visualization of the activated pathways.
> The distribution of expert loading examines the expert use frequency for all tokens.
> Activated pathways examine the behavior of experts at the token level: this visualization tool tracts the activated pathways of all tokens on validation datasets; given all activated pathways, the visualization tool employs PCA to obtain the top-10 pathways.
>
> As shown in Figure 5 of the manuscript, we find that
> $(i)$ STGC benefits the expert load balance.
> A possible reason is that the expert load imbalance means that many tokens are routed to an expert, significantly increasing the possibility that tokens have gradient conflicts.
> After adding the STGC, some tokens are moved from the "crowded" expert (many tokens) to the "empty" expert (few tokens).
> This validates that STGC would effectively utilize each expert, instead of collapsing into only using one expert.
> $(ii)$ The activated pathways are significantly different for SQA, TextVQA, and MMBench.
> This implies that although the distribution of expert load across different datasets is similar, the token routing behavior is still significantly different among datasets, \textit{i.e.}, different tokens have been assigned to various experts.
>
> > _Comment 2: It seems that the optimization process of the proposed CEL is not very stable (as in Figure 3(a)), the reviewer wonders whether this would be a problem when scaling up to larger scale._
>
> Response: The CEL appears unstable possibly because too few steps are displayed. We present the actual CEL curve during the training process (5197 steps) in **Figure 6** of the revised manuscript and find that the CEL is convergent.

---

> > ### Author Response · Authors · 2024-11-23
> > **Response to Reviewer EECo (Part 2)**
> >
> > > _Comment 3: Computational overhead should be provided to ensure the proposed method does not incur too much overhead. This added study can show whether the proposed method is promising in terms of scaling up to large scale._
> >
> > Response: Thanks for your advice.
> > We have conducted a thorough computational overhead analysis in **Section 6.6** and **Table 10** of the revised manuscript.
> >
> > **Training memory overhead**.
> > We compute the memory overhead by the size of the stored gradient.
> > As stated in Section 6.1, for each token, we only store the gradient it produces on the bias within the experts. Thus, the theoretical memory overhead is merely about 0.29 GB on StableLM-1.6B and 0.38 GB on Phi2-2.7B, which is negligible during the training of MoE-based LVLMs.
> > For the specific calculation please refer to Section 6.6.
> >
> > **Training time overhead**.
> > We directly report the train_samples_per_second and train_steps_per_second recorded in "trainer_state.json" after training.
> > MoE-LLaVA performs a forward pass, followed by a backward pass to update parameters.
> > As Section 6.1 mentioned, MoE-LLaVA+STGC freezes parameters except for the bias within the experts, performs a forward pass, followed by a backward pass to compute token-level gradients; then, MoE-LLaVA+STGC unfreeze parameters and performs a backward pass to update parameters.
> > Thus, the main time overhead results from "the computation of token-level gradients".
> > As shown in the below tables, we have reduced the additional time overhead to about 20\% through some engineering tricks (_e.g._, only computing the token-level gradient on the bias and freezing parameters that do not require gradient computation).
> > Some other methods may further speed up STGC, such as using STGC only on even iterations or applying STGC to only half of the MoE layers.
> > We will further explore these experiments in the future.
> >
> > Computational overhead analysis on StableLM-1.6B:
> > | Method      |  (theoretical memory overhead) | (train_samples_per_second) | (train_steps_per_second) | (one step (s))       |
> > |-------------|-----------------------------|--------------------------|------------------------|--------------------|
> > | MoE-LLaVA   | -                           | 19.796                   | 0.154                  | 6.494              |
> > | +STGC       | 0.29 GB                     | 16.283                   | 0.127                  | 7.874 (+21.3%)     |
> > | +STGC-full  | 773 GB                    | fail                     | fail                   | fail               |
> >
> > (_STGC-full means computing token-level gradients on the weight.
> > Meanwhile, STGC is the used scheme, only computing token-level gradients on the bias._)
> >
> > Computational overhead analysis on Phi2-2.7B:
> > | Method      | (theoretical memory overhead) | (train_samples_per_second) | (train_steps_per_second) | (one step (s))       |
> > |-------------|-------------------------------|-----------------------------|---------------------------|----------------------|
> > | MoE-LLaVA   | -                             | 10.765                      | 0.084                     | 11.905               |
> > | +STGC       | 0.38 GB                       | 8.851                       | 0.069                     | 14.493 (+21.7%)      |
> > | +STGC-full  | 1562 GB                       | fail                        | fail                      | fail                 |

---

> > > ### Author Response · Authors · 2024-11-28
> > > **Response to Reviewer EECo**
> > >
> > > We further implement the experiment on a larger scale of LLM in **Section 6.12** and **Table 15**.
> > >
> > > Specifically, we expand **Vicuna-7B-v1.5** (V-7B, the LLM used in LLaVA-1.5) into MoE.
> > > Like the MoE design applied to StableLM-1.6B and Phi2-2.7B, we duplicate the FFN in Vicuna-7B four times to form an MoE layer. During training and inference, we select the Top-2 experts to process tokens, _i.e._, the 4Top2 configuration.
> > > Given the large size of the model (potentially exceeding 20 billion parameters), we employ distributed parameter deployment across multiple GPUs for training. Meanwhile, the batch size is reduced to 1, and gradient accumulation is set to 16.
> > > As shown in the table below, we implement MoE-LLaVA and MoE-LLaVA+STGC and report their performance. The conclusions drawn from Vicuna-7B-v1.5 are consistent with those drawn from StableLM-1.6B and Phi2-2.7B.
> > >
> > > | Method              | LLM    | VQA$^\text{v2}$ | GQA  | VisWiz | SQA$^\text{I}$ | VQA$^\text{T}$ | POPE  | MME   | MMB  | MM-Vet | Avg  |
> > > |---------------------|--------|-----------------|------|--------|----------------|----------------|-------|-------|------|--------|------|
> > > | MoE-LLaVA-4Top2    | V-7B   | 77.0$^*$        | 61.1$^*$ | 51.1  | 69.3           | 54.8           | 86.0 | 1366.9| 67.1 | 29.8   | 62.0 |
> > > | +STGC               | V-7B   | 77.3$^*$        | 61.2$^*$ | 51.9  | 71.9           | 55.3           | 86.1 | 1409.3| 67.4 | 31.6   | 62.8 |

---

### Official Review · Reviewer_U9YJ · 2024-11-04

**Soundness:** 3
**Presentation:** 3
**Contribution:** 3
**Rating:** 6
**Confidence:** 3

**Summary:**

This paper introduces an approach to improve the Mixture-of-Expert(MOE) by eliminating token-level gradient conflict during training. The gradient of the token is compared with the average gradient to decide the conflict tokens. A Conflict Elimination Loss (CEL) is proposed as a regularization term to encourage the reassignment of conflicting tokens. The results on various model setting and tasks validate the effectiveness of proposed method.

**Strengths:**

- The paper is well-written and easy to follow. The motivation for solving token-level gradient conflict during training MOE is sound to me.
- The proposed CEL can serve as a direct plug-in to improve the training of MOEs.
- A consistent improvement can be observed in Table 2 and 3, and the proposed STGC can be applicable to a wide range of model and data settings.
- The evaluation of gradient consistency in Figure 3 also provides further evidence to account for the improvement.

**Weaknesses:**

- It would be better to illustrate the distribution of cosine similarity between gradients of all tokens and the averaged gradient for better understanding.
- Since the average gradient is dynamically changing during training, is it possible that some tokens do not conflict with a specific expert but become conflicting after several steps of training?
- Why only the Large Vision-Language Model (LVLM) setting is considered?

- Some minor points:
  - Figure 3b: better label the x-axis with "Training Step"
  - In TL;DR, a novel loss.
  - Figure A in supplementary: part of the figure is not visible

**Questions:**

- Will it introduce additional training costs for the computation of the gradient and similarity?
- It would be better to provide some analysis on the distribution of expert loadings for better understanding.

---

> ### Author Response · Authors · 2024-11-23
> **Response to Reviewer U9YJ (Part 1)**
>
> Thank you for your review and your feedback on our paper. We address your questions and concerns below.
>
> > _Comment 1: It would be better to illustrate the distribution of cosine similarity between gradients of all tokens and the averaged gradient for better understanding._
>
> Response: Thanks for your advice.
> We have supplemented the distribution of cosine similarity between the gradient of each token and the averaged gradient in **Section 6.5** and **Figure 7** of the revised manuscript.
> We find that STGC can effectively increase the cosine similarity between the gradients of tokens within an expert and the average gradient.
>
> > _Comment 2: Since the average gradient is dynamically changing during training, is it possible that some tokens do not conflict with a specific expert but become conflicting after several steps of training?_
>
> Response: It is possible that _conflicting tokens_ are dynamic.
> Hence, rather than identifying _conflicting tokens_ before training, STGC dynamically identifies _conflicting tokens_ during training and constrains their routing through conflict elimination loss.
> Experimental results in Figure 7 and Figure 8 of the manuscript have indicated that STGC effectively reduces the number of _conflicting tokens_.
>
> > _Comment 3: Why only the Large Vision-Language Model (LVLM) setting is considered?_
>
> Response: Due to limited time, we did not apply STGC to other tasks in the original manuscript.
> Theoretically, the deployment of STGC is not restricted to task types.
> To verify this, we extend STGC to language tasks.
> We are still preparing the experimental results of this part, and we will report them within the next day or two.
>
> > _Comment 4: Some minor points._
>
> Response: Thanks for your advice.
> We have fixed all the points you mentioned except for TL;DR as we cannot change it now.

---

> > ### Author Response · Authors · 2024-11-23
> > **Response to Reviewer U9YJ (Part 2)**
> >
> > > _Comment 5: Will it introduce additional training costs for the computation of the gradient and similarity?_
> >
> > Response: Yes.
> > We have conducted a thorough computational overhead analysis in **Section 6.6** and **Table 10** of the revised manuscript.
> >
> > **Training memory overhead**.
> > We compute the memory overhead by the size of the stored gradient.
> > As stated in Section 6.1, for each token, we only store the gradient it produces on the bias within the experts. Thus, the theoretical memory overhead is merely about 0.29 GB on StableLM-1.6B and 0.38 GB on Phi2-2.7B, which is negligible during the training of MoE-based LVLMs.
> > For the specific calculation please refer to Section 6.6.
> >
> > **Training time overhead**.
> > We directly report the train_samples_per_second and train_steps_per_second recorded in "trainer_state.json" after training.
> > MoE-LLaVA performs a forward pass, followed by a backward pass to update parameters.
> > As Section 6.1 mentioned, MoE-LLaVA+STGC freezes parameters except for the bias within the experts, performs a forward pass, followed by a backward pass to compute token-level gradients; then, MoE-LLaVA+STGC unfreeze parameters and performs a backward pass to update parameters.
> > Thus, the main time overhead results from "the computation of token-level gradients".
> > As shown in the below tables, we have reduced the additional time overhead to about 20\% through some engineering tricks (_e.g._, only computing the token-level gradient on the bias and freezing parameters that do not require gradient computation).
> > Some other methods may further speed up STGC, such as using STGC only on even iterations or applying STGC to only half of the MoE layers.
> > We will further explore these experiments in the future.
> >
> > Computational overhead analysis on StableLM-1.6B:
> > | Method      |  (theoretical memory overhead) | (train_samples_per_second) | (train_steps_per_second) | (one step (s))       |
> > |-------------|-----------------------------|--------------------------|------------------------|--------------------|
> > | MoE-LLaVA   | -                           | 19.796                   | 0.154                  | 6.494              |
> > | +STGC       | 0.29 GB                     | 16.283                   | 0.127                  | 7.874 (+21.3%)     |
> > | +STGC-full  | 773 GB                    | fail                     | fail                   | fail               |
> >
> > (_STGC-full means computing token-level gradients on the weight.
> > Meanwhile, STGC is the used scheme, only computing token-level gradients on the bias._)
> >
> > Computational overhead analysis on Phi2-2.7B:
> > | Method      | (theoretical memory overhead) | (train_samples_per_second) | (train_steps_per_second) | (one step (s))       |
> > |-------------|-------------------------------|-----------------------------|---------------------------|----------------------|
> > | MoE-LLaVA   | -                             | 10.765                      | 0.084                     | 11.905               |
> > | +STGC       | 0.38 GB                       | 8.851                       | 0.069                     | 14.493 (+21.7%)      |
> > | +STGC-full  | 1562 GB                       | fail                        | fail                      | fail                 |
> >
> > > _Comment 6: It would be better to provide some analysis on the distribution of expert loadings for better understanding._
> >
> > Response: Thanks for your advice.
> > We have presented the distribution of expert loading in **Section 6.3** and **Figure 5** of the revised manuscript.
> > In specific, we follow MoE-LLaVA to compute the distribution of expert loading.
> > The distribution of expert loading examines the expert use frequency for all tokens.
> > As shown in Figure 5 in the manuscript, we find that STGC benefits the expert load balance.
> > A possible reason is that the expert load imbalance means that many tokens are routed to an expert, significantly increasing the possibility that tokens have gradient conflicts.
> > After adding the STGC, some tokens are moved from the "crowded" expert (many tokens) to the "empty" expert (few tokens).
> > This may become a new perspective on why the load balance is important to the MoE system.

---

> > > ### Comment · Reviewer_U9YJ · 2024-11-25
> > >
> > > Thanks for the detailed feedback. Most of my concerns are addressed, and I'd like to maintain my score.

---

> > > > ### Author Response · Authors · 2024-11-25
> > > > **Response to Reviewer U9YJ**
> > > >
> > > > Thanks for your recognition of our response! We sincerely appreciate the time and effort you've dedicated to our paper.
> > > >
> > > > We would like to update the experimental result below. Sorry for the late.
> > > > > _Comment 3: Why only the Large Vision-Language Model (LVLM) setting is considered?_
> > > >
> > > > Response: Due to limited time, we did not apply STGC to other tasks in the original manuscript.
> > > > Theoretically, the deployment of STGC is not constrained to specific task types.
> > > > Thus, to validate the generalization of STGC, we extend its use to language tasks.
> > > > We have supplemented the study on language tasks on **Section 6.10** and **Table 13**.
> > > > We follow DYNMOE [1] to apply the MoE framework for language tasks and we add the proposed STGC to the MoE.
> > > > As shown in the below table, the experimental results show the significant effectiveness of STGC on language tasks.
> > > > For more details please refer to Section 6.10 and Table 13.
> > > >
> > > > |                |  COLA  |  MRPC  |  QNLI  |  MNLI  |  RTE   |  Avg  |
> > > > |-----------------|--------|--------|--------|--------|--------|-------|
> > > > | MoE-8Top2 [1] |  64.5  |  90.2  |  92.4  |  86.7  |  74.9  | 81.7 |
> > > > | DYNMOE [1]   |  65.2  |  90.6  |  92.6  |  86.4  |  73.4  | 81.6 |
> > > > | MoE-8Top2$^*$   |  64.5  |  90.0  |  93.4  |  86.9  |  72.9  | 81.5 |
> > > > | +STGC           |  66.8  |  91.2  |  93.8  |  87.6  |  74.7  | 82.8 |
> > > >
> > > > $*$ means the re-implemented results.
> > > >
> > > > [1] Guo, Yongxin, et al. "Dynamic mixture of experts: An auto-tuning approach for efficient transformer models." arXiv preprint arXiv:2405.14297 (2024).

---

### Author Response · Authors · 2024-12-02
**Grateful Response to all Reviewers**

Dear Reviewers,

We sincerely thank you for your thoughtful comments and constructive suggestions, which have helped strengthen our paper. We have submitted our responses and the revised manuscript with changes highlighted in blue. As the discussion phase is coming to a close, we kindly invite you to review our responses. We hope they effectively address your concerns. If you have any further questions or concerns, please feel free to reach out.

Thank you once more for your support and collaboration!

Best regards,

The Authors

---

### Meta-Review · Area_Chair_7oUd · 2024-12-21

**Metareview:**

This paper proposes a method to reduce the interference between tokens in Mixture-of-Experts models. This is done by adding a loss term that aligns the per-expert token gradients to the mean gradient over all tokens within the expert. The authors show that this proecedure does indeed improve the alignment of the token gradients, and that this leads to better performance during training.

Reviewers were initially positive about the paper, and the comprehensive rebuttal further addressed the remaining concerns. Therefore, the decision is to accept the paper.

**Additional Comments On Reviewer Discussion:**

Please see above. Reviewers were initially positive about the paper, and the comprehensive rebuttal further addressed the remaining concerns.

---

### Decision · Program_Chairs · 2025-01-22

Accept (Poster)